# Bounds on Representation-Induced Confounding Bias for Treatment Effect Estimation

**Valentyn Melnychuk, Dennis Frauen & Stefan Feuerriegel**
LMU Munich & Munich Center for Machine Learning
Munich, Germany
`melnychuk@lmu.de`

## Abstract

State-of-the-art methods for conditional average treatment effect (CATE) estimation make widespread use of representation learning. Here, the idea is to reduce the variance of the low-sample CATE estimation by a (potentially constrained) low-dimensional representation. However, low-dimensional representations can lose information about the observed confounders and thus lead to bias, because of which the validity of representation learning for CATE estimation is typically violated. In this paper, we propose a new, representation-agnostic refutation framework for estimating bounds on the *representation-induced confounding bias* that comes from dimensionality reduction (or other constraints on the representations) in CATE estimation. First, we establish theoretically under which conditions CATE is non-identifiable given low-dimensional (constrained) representations. Second, as our remedy, we propose a neural refutation framework which performs partial identification of CATE or, equivalently, aims at estimating lower and upper bounds of the representation-induced confounding bias. We demonstrate the effectiveness of our bounds in a series of experiments. In sum, our refutation framework is of direct relevance in practice where the validity of CATE estimation is of importance.

## 1 Introduction

Estimating conditional average treatment effect (CATE) from observational data is important for many applications in medicine (Kraus et al., 2023; Feuerriegel et al., 2024), marketing (Varian, 2016), and economics (Imbens & Angrist, 1994). For example, medical professionals use electronic health records to personalize care based on the estimated CATE.

Different machine learning methods have been developed for CATE estimation (see Sec. 2 for an overview). In this paper, we focus on representation learning methods (e.g., Johansson et al., 2016; Shalit et al., 2017; Hassanpour & Greiner, 2019b;a; Zhang et al., 2020; Assaad et al., 2021; Johansson et al., 2022). Representation learning methods have several benefits: (1) They often achieve state-of-the-art performance, especially in low-sample regimens. (2) They provide generalization bounds for best-in-class estimation, regardless of whether ground-truth CATE belongs to a specified model class. (3) They manage to reduce the variance of the low-sample CATE estimation by using a (potentially constrained) low-dimensional representation. Often, constraints are imposed on the representations such as balancing with an empirical probability metric and invertibility.

While representation learning methods benefit from reducing variance, they also have a shortcoming: low-dimensional (potentially constrained) representations can lose information about covariates, including information about ground-truth confounders. As we show later, such low-dimensional representations can thus lead to bias, because of which the validity of representation learning methods may be violated. To this end, we introduce the notion of *representation-induced confounding bias* (RICB). As a result of the RICB, the validity of representation learning for CATE estimation is typically violated, and we thus offer remedies in our paper.

In this paper, we propose a new, representation-agnostic refutation framework for estimating bounds on the RICB that comes from dimensionality reduction (or other constraints on the representation). First, we show in which settings CATE is non-identifiable due to the RICB. We further discuss how

Table 1: Overview of key representation learning methods for CATE estimation.

| Method | Invertibility | Balancing with | |
| --- | --- | --- | --- |
| | | empirical probability metrics | loss re-weighting |
| TARNet (Shalit et al., 2017; Johansson et al., 2022) | – | – | – |
| BNN (Johansson et al., 2016); CFR (Shalit et al., 2017; Johansson et al., 2022); ESCFR (Wang et al., 2024) | – | IPM (MMD, WM) | – |
| RCFR (Johansson et al., 2018; 2022) | – | IPM (MMD, WM) | Learnable weights |
| DACPOL (Atan et al., 2018); CRN (Bica et al., 2020); ABCEI (Du et al., 2021); CT (Melnychuk et al., 2022); MitNet (Guo et al., 2023); BNCDE (Hess et al., 2024) | – | JSD (adversarial learning) | – |
| SITE (Yao et al., 2018) | Local similarity | Middle point distance | – |
| CFR-ISW (Hassanpour & Greiner, 2019a); DR-CFR (Hassanpour & Greiner, 2019b); DeR-CFR (Wu et al., 2022) | – | IPM (MMD, WM) | Representation propensity |
| DKLITE (Zhang et al., 2020) | Reconstruction loss | Counterfactual variance | – |
| BWCFR (Assaad et al., 2021) | – | IPM (MMD, WM) | Covariate propensity |
| PM (Schwab et al., 2018); StableCFR (Wu et al., 2023) | – | – | Upsampling via matching |

IPM: integral probability metric; MMD: maximum mean discrepancy; WM: Wasserstein metric; JSD: Jensen-Shannon divergence

different constraints imposed on the representation impact the RICB. Then, as a remedy, we perform partial identification of CATE or, equivalently, estimate lower and upper bounds on the RICB.

We empirically demonstrate the effectiveness of our refutation framework on top of many state-of-the-art representation learning methods for CATE estimation. We thereby show that the established representation learning methods for CATE estimation such as BNN (Johansson et al., 2016); TARNet, CFR, RCFR (Shalit et al., 2017; Johansson et al., 2018; 2022); CFR-ISW (Hassanpour & Greiner, 2019a); and BWCFR (Assaad et al., 2021) can be more reliable for decision-making when combined with our refutation framework. We thus evaluate decision-making based on CATE, finding that the policies with deferral that account for our bounds on the RICB achieve lower error rates than the decisions based on CATE estimates from the original representation learning method. As such, our refutation framework offers a tool for practitioners to check the validity of CATE estimates from representation learning methods and further improve the reliability and safety of representation learning in CATE estimation.

In sum, our **contributions** are following:[1] (1) We show that the CATE from representation learning methods can be non-identifiable due to a *representation-induced confounding bias*. To the best of our knowledge, we are the first to formalize such bias. (2) We propose a representation-agnostic refutation framework to perform partial identification of CATE, so that we estimate lower and upper bounds of the representation-induced confounding bias. (3) We demonstrate the effectiveness of our bounds together with a wide range of state-of-the-art CATE methods.

## 2 RELATED WORK

An extensive body of literature has focused on CATE estimation. We provide an extended overview in Appendix A, while, in the following, we focus on two important streams relevant to our framework.

**Representation learning for CATE estimation.** Seminal works of Johansson et al. (2016); Shalit et al. (2017); and Johansson et al. (2022) provided generalization bounds for representation learning methods aimed at CATE estimation under the assumption of *invertibility* of the representations. Therein, the authors demonstrated, that generalization bounds depend on the imbalance of the invertible representations between treated and untreated subgroups, and some form of *balancing* is beneficial for reducing the variance. However, although invertibility of representations is still required theoretically, it is usually not enforced in practice (see Table 1). Namely, due to bias-variance trade-off, low-dimensional (non-invertible) representations can still generalize better, although potentially containing confounding bias (Ding et al., 2017).

Numerous works offer a variety of tailored deep representation learning methods for CATE estimation by extending TARNet (Shalit et al., 2017; Johansson et al., 2022) (the simplest representation leaning CATE estimator) with different ways to enforce (i) invertibility and (ii) balancing. For example, (i) invertibility was enforced via local similarity in SITE (Yao et al., 2018), and via reconstruction loss in DKLITE (Zhang et al., 2020). (ii) Balancing was implemented as both (1) a constraint on the representation with some empirical probability metric and (2) loss re-weighting. Examples of (1) include balancing with *integral probability metrics* in, e. g., BNN (Johansson et al., 2016), and CFR (Shalit et al., 2017; Johansson et al., 2022); *Jensen-Shannon divergence* in, e. g., DACPOL (Atan et al., 2018), CT (Melnychuk et al., 2022), and MitNet (Guo et al., 2023); *middle point distance*

---

[1]Code is available at `https://github.com/Valentyn1997/RICB`.

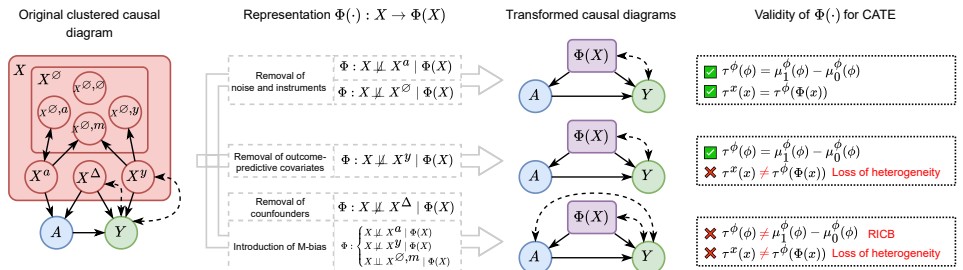

Figure 1: The validity of CATE estimation is influenced by the different constraints imposed on representations $\Phi(\cdot)$. In red: different violations of valid CATE estimation.

in SITE (Yao et al., 2018); and *counterfactual variance* in DKLITE (Zhang et al., 2020). (2) Loss re-weighting was implemented with *learnable weights* in RCFR (Johansson et al., 2018; 2022); *inverse propensity weights* in, e. g., CFR-ISW (Hassanpour & Greiner, 2019a) and BWCFR (Assaad et al., 2021); and *upsampling* in PM (Schwab et al., 2018) and StableCFR (Wu et al., 2023). We provide the full overview of key representation learning methods for CATE in Table 1.

**Handling unobserved confounding.** Sensitivity models are used to estimate the bias in treatment effect estimation due to the *hidden confounding* by limiting the strength of hidden confounding through a sensitivity parameter. There are two main classes of sensitivity models: outcome sensitivity models (OSMs) (Robins et al., 2000; Blackwell, 2014; Bonvini et al., 2022) and propensity (marginal) sensitivity models (MSMs) (Tan, 2006; Jesson et al., 2021; Bonvini et al., 2022; Dorn & Guo, 2022; Frauen et al., 2023; Oprescu et al., 2023). We later adopt ideas from the MSMs to derive our refutation framework, because they only require knowledge of propensity scores wrt. covariates and representation, but *not* the actual expected potential outcomes as required by OSMs.

Of note, our method uses MSMs but in an unconventional way. Importantly, we do not use MSMs for sensitive analysis but for partial identification. As such, we do *not* face the common limitation of MSMs in that the sensitivity parameter (which guides the amount of hidden confounding) must be chosen through expert knowledge. In contrast, our application allows us to estimate the sensitivity parameters in the MSM from data.

**Research gap:** To the best of our knowledge, no work has studied the confounding bias in low-dimensional (constrained) representations for CATE estimation. Our novelty is to formalize the *representation-induced confounding bias* and to propose a neural refutation framework for estimating bounds.

## 3 VALIDITY OF REPRESENTATION LEARNING FOR CATE

In the following, we first formalize representation learning for CATE estimation (Sec. 3.1). We then define the concept of valid representations and give two conditions when this is violated (Sec. 3.2). Finally, we lay out the implications of invalid representations (Sec. 3.3).

**Notation.** Let $X$ be a random variable with a realization $x$, distribution $\mathbb{P}(X)$, density/probability function $\mathbb{P}(X = x)$ and domain $\mathcal{X}$, respectively. Furthermore, let $\mathbb{P}(Y \mid X = x, A = a)$ be the conditional distribution of the outcome $Y$. Let $\pi_a^x(x) = \mathbb{P}(A = a \mid X = x)$ denote the covariate propensity score for treatment $A$ and covariates $X$, and $\mu_a^x(x) = \mathbb{E}(Y \mid A = a, X = x)$ an expected covariate-conditional outcome. Analogously, $\pi_a^\phi(\phi) = \mathbb{P}(A = a \mid \Phi(X) = \phi)$ and $\mu_a^\phi(\phi) = \mathbb{E}(Y \mid A = a, \Phi(X) = \phi)$ are the representation propensity score and the expected representation-conditional outcome, respectively, for treatment $A$ and representation $\Phi(X)$. Importantly, in the definitions of $\pi_a^x, \mu_a^x, \pi_a^\phi$, and $\mu_a^\phi$, upper indices serve as indicators that the nuisance functions relate either to the covariates or to the representations and, therefore, are not arguments. Let $Y[a]$ denote a potential outcome after intervening on the treatment $do(A = a)$. $\text{dist}[\cdot]$ is some distributional distance.

### 3.1 REPRESENTATION LEARNING FOR CATE

**Problem setup.** Assume we have observational data $\mathcal{D}$ with a binary treatment $A \in \{0, 1\}$, high-dimensional covariates $X \in \mathcal{X} \subseteq \mathbb{R}^{d_x}$, and a continuous outcome $Y \in \mathcal{Y} \subseteq \mathbb{R}$. In medicine, $A$ could be ventilation, $X$ the patient's health history, and $Y$ respiratory rate.

Without a loss of generality, we assume that covariates $X$ are an implicit cluster (Anand et al., 2023) of four sub-covariates, $X = \{X^\varnothing, X^a, X^y, X^\Delta\}$, namely, (1) noise, (2) instruments, (3) outcome-predictive covariates, and (4) confounders (Cinelli et al., 2022) (see clustered causal diagram in Fig. 1). The noise, in turn, can be partitioned onto (1.1) an independent noise, (1.2) descendants of instruments, (1.3) descendants of outcome-predictive covariates, and (1.4) M-bias-inducing covariates, namely, $X^\varnothing = \{X^{\varnothing,\varnothing}, X^{\varnothing,a}, X^{\varnothing,y}, X^{\varnothing,m}\}$. Importantly, some sub-covariates could be empty, and the partitioning of $X$ is usually unknown in practice, and, thus, it will be only used in this paper to provide better intuition (all experiments later consider the partitioning as unknown). The observational data are sampled i.i.d. from a joint distribution, i. e., $\mathcal{D} = \{x_i, a_i, y_i\}_{i=1}^n \sim \mathbb{P}(X, A, Y)$, where $n$ is the sample size. Furthermore, potential outcomes are only observed for factual treatments, i. e., $Y = A\, Y[1] + (1 - A)\, Y[0]$, which is referred as the fundamental problem of causal inference.

The *conditional average treatment effect* (CATE) is then defined as

$$\tau^x(x) = \mathbb{E}(Y[1] - Y[0] \mid X = x), \tag{1}$$

where upper index $\tau$ indicates that CATE is with respect to the covariates $x$. Identification of CATE from observational data relies on Neyman–Rubin potential outcomes framework (Rubin, 1974). As such, we assume (i) *consistency:* if $A = a$ then $Y = Y[a]$; (ii) *overlap:* $\mathbb{P}(0 < \pi_a^x(X) < 1) = 1$; and (iii) *exchangeability:* $A \perp\!\!\!\perp (Y[0], Y[1]) \mid X$. Under assumptions (i)–(iii), the CATE is **identifiable** from observational data, $\mathbb{P}(X, A, Y)$, i. e. from expected covariate-conditional outcomes, $\tau^x(x) = \mu_1^x(x) - \mu_0^x(x)$.

**Representation learning for CATE estimation.** Representations learning methods CATE estimators (Johansson et al., 2016; Shalit et al., 2017; Johansson et al., 2022) do not assume a specific partitioning of covariates $X$ and generally consist of two main components: (1) the *representation subnetwork* $\Phi(\cdot)$ and (2) the *potential outcomes predicting subnetwork(s)*, i. e., $f_0(\cdot), f_1(\cdot)$.[2] Both components are as follows. (1) The representation subnetwork maps all the covariates $X \to \Phi(X)$ to a low- or equal-dimensional representation space with some measurable function, $\Phi(\cdot) : \mathbb{R}^{d_x} \to \mathbb{R}^{d_\phi}, d_\phi \leq d_x$.[3] Additional constraints can be imposed on $\Phi(\cdot)$. For example, balancing with an empirical probability metric, $\mathrm{dist}\left[\mathbb{P}(\Phi(X) \mid A = 0), \mathbb{P}(\Phi(X) \mid A = 1)\right] \approx 0$, and invertibility, $\Phi^{-1}(\Phi(X)) \approx X$. (2) The potential outcomes predicting subnetwork(s) then aim at estimating CATE with respect to the representations, namely

$$\tau^\phi(\phi) = \mathbb{E}(Y[1] - Y[0] \mid \Phi(X) = \phi), \tag{2}$$

where $\phi = \Phi(x)$. Yet, $f_0$ and $f_1$ can only access representation-conditional outcomes, and, therefore, instead, estimate $\mu_0^\phi(\phi)$ and $\mu_1^\phi(\phi)$, respectively.

## 3.2 Valid representations

In the following, we discuss under what conditions representations $\Phi(\cdot)$ are valid for CATE estimation, namely, do not introduce an infinite-data bias.

**Definition 1** (Valid representations). *We call a representation $\Phi(\cdot)$ valid for CATE if it satisfies the following two equalities:*

$$\tau^x(x) \overset{(i)}{=} \tau^\phi(\Phi(x)) \quad and \quad \tau^\phi(\phi) \overset{(ii)}{=} \mu_1^\phi(\phi) - \mu_0^\phi(\phi), \tag{3}$$

*where $\tau^x(\cdot)$ and $\tau^\phi(\cdot)$ are CATEs wrt. covariates and representations from Eq. (1) and (2), respectively.*

If equality $(i)$ is violated, then we say that the representation suffers from a *loss in heterogeneity*. If $(ii)$ is violated, it suffers from a *representation-induced confounding bias*. Hence, if any of the two is violated, we have invalid representations (see Fig. 1, left).

**Characterization of valid representations.** We now give two examples of valid representations based on whether information about some sub-covariates $\{X^\varnothing, X^a, X^y, X^\Delta\}$ is fully preserved in $\Phi(X)$.

---

[2] The latter can have either one subnetwork with two outputs (SNet) as in, e. g., BNN (Johansson et al., 2016); or two subnetworks (TNet) as in, e. g., TARNet (Shalit et al., 2017).

[3] The transformation $\Phi(\cdot)$ can also be seen as a mapping between micro-level covariates $X$ to low-dimensional macro-level representations (Rubenstein et al., 2017).

■ *Invertible representations.* A trivial example of a valid representation is an invertible representation (Shalit et al., 2017; Zhang et al., 2020; Johansson et al., 2022):

$$\tau^x(x) \stackrel{(i)}{=} \mathbb{E}(Y[1] - Y[0] \mid X = \Phi^{-1}(\Phi(x))) = \mathbb{E}(Y[1] - Y[0] \mid \Phi(X) = \Phi(x)) = \tau^\phi(\Phi(x)),$$

$$\tau^x(x) \stackrel{(ii)}{=} \mathbb{E}\left(Y \mid A = 1, X = \Phi^{-1}(\Phi(x))\right) - \mathbb{E}\left(Y \mid A = 0, X = \Phi^{-1}(\Phi(x))\right) = \mu_1^\phi(\Phi(x)) - \mu_0^\phi(\Phi(x)),$$

where equality $(ii)$ follows from setting $\Phi(x)$ to $\phi$. Notably, if $\Phi(\cdot)$ is an invertible transformation, then

$$X \perp\!\!\!\perp X^\varnothing \mid \Phi(X), \quad X \perp\!\!\!\perp X^a \mid \Phi(X), \quad X \perp\!\!\!\perp X^y \mid \Phi(X), \quad X \perp\!\!\!\perp X^\Delta \mid \Phi(X), \tag{4}$$

i. e., there is no loss of information on each of the sub-covariates. Eq. (4) holds, as conditioning on $\Phi(X)$ renders each sub-covariate deterministic, thus implying independence. In Lemma 1 of Appendix B, we also show that the opposite statement is true: when all the statements hold in Eq. (4), then $\Phi(\cdot)$ is an invertible transformation.

■ *Removal of noise and instruments.* Another class of valid representations are those, which lose any amount of information about the noise, $X^\varnothing$, or the instruments, $X^a$:

$$X \not\perp\!\!\!\perp X^\varnothing \mid \Phi(X) \quad \text{or} \quad X \not\perp\!\!\!\perp X^a \mid \Phi(X). \tag{5}$$

The validity follows from the d-separation in the clustered causal diagram (Fig. 1) and invertibility of $\Phi(\cdot)$ wrt. $X^\Delta$ and $X^y$ (see Lemma 2 in Appendix B):

$$\tau^x(x) \stackrel{(i)}{=} \mathbb{E}(Y[1] - Y[0] \mid x) = \mathbb{E}(Y[1] - Y[0] \mid x^\Delta, x^y) = \mathbb{E}(Y[1] - Y[0] \mid \Phi(x)) = \tau^\phi(\Phi(x)),$$

$$\tau^x(x) \stackrel{(ii)}{=} \mathbb{E}\left(Y \mid A = 1, x^\Delta, x^y\right) - \mathbb{E}\left(Y \mid A = 0, x^\Delta, x^y\right) \tag{6}$$

$$= \mathbb{E}\left(Y \mid A = 1, \Phi(x)\right) - \mathbb{E}\left(Y \mid A = 0, \Phi(x)\right) = \mu_1^\phi(\Phi(x)) - \mu_0^\phi(\Phi(x)),$$

where equality $(ii)$ follows from setting $\Phi(x)$ to $\phi$.

In actual implementations, representation learning methods for CATE achieve (1) the loss of information about the instruments through balancing with an empirical probability metric, and (2) the loss of information about the noise by lowering the representation size (which is enforced with the factual outcome loss).

## 3.3 Implications of Invalid representations

In the following, we discuss the implications for CATE estimation from (1) loss of heterogeneity and (2) RICB for CATE estimation, and in what scenarios they appear in low-dimensional or balanced representations.

$(i)$ **Loss of heterogeneity.** The loss of heterogeneity happens whenever $\tau^x(x) \neq \tau^\phi(\Phi(x))$. It means that the treatment effect at the covariate (individual) level is different from the treatment effect at the representation (aggregated) level. As an example, such discrepancy can occur due to aggregation such as a one-dimensional representation (e.g., as in the covariate propensity score, $\Phi(X) = \pi_1^x(X)$). In this case, the treatment effect $\tau^\phi(\phi)$ denotes a propensity conditional average treatment effect, which is used in propensity matching.

The loss of heterogeneity happens whenever some information about $X^\Delta$ or $X^y$ is lost in the representation, i. e.,

$$X \not\perp\!\!\!\perp X^\Delta \mid \Phi(X) \quad \text{or} \quad X \not\perp\!\!\!\perp X^y \mid \Phi(X). \tag{7}$$

This could be the case due to a too low dimensionality of the representation so that the information on $X^y$ or $X^\Delta$ is lost, or due to too large balancing with an empirical probability metric so that we lose $X^\Delta$. Then, the equality $(i)$ does not any longer hold in Eq. (3), i. e.,

$$\tau^\phi(\Phi(x)) \stackrel{(i)}{=} \mathbb{E}(Y[1] - Y[0] \mid \Phi(x)) = \int_{\mathcal{X}_\Delta \times \mathcal{X}_Y} \mathbb{E}(Y[1] - Y[0] \mid \Phi(x), x^\Delta, x^y) \, \mathbb{P}(X^\Delta = x^\Delta, X^y = x^y \mid \Phi(x)) \, \mathrm{d}x^\Delta \, \mathrm{d}x^y$$

$$= \int_{\mathcal{X}_\Delta \times \mathcal{X}_Y} \tau^x(x) \, \mathbb{P}(X^\Delta = x^\Delta, X^y = x^y \mid \Phi(x)) \, \mathrm{d}x^\Delta \, \mathrm{d}x^y \neq \tau^x(x).$$

Although the CATE wrt. representations, $\tau^\phi(\phi)$, and the CATE wrt. covariates, $\tau^x(x)$, are different, the CATE wrt. representations is identifiable from observational data $\mathbb{P}(\Phi(X), A, Y)$ due to the exchangeability wrt. representations, $A \perp\!\!\!\perp (Y[0], Y[1]) \mid \Phi(X)$. This is seen in the following:

$$\tau^\phi(\phi) \stackrel{(ii)}{=} \mathbb{E}(Y[1] - Y[0] \mid \phi) = \mathbb{E}(Y[1] \mid A = 1, \phi) - \mathbb{E}(Y[0] \mid A = 0, \phi) = \mu_1^\phi(\phi) - \mu_0^\phi(\phi). \tag{8}$$

($ii$) **Representation-induced confounding bias (RICB).** This situation happens when information about $X^\Delta$ is lost or when M-bias is introduced, i.e., some information is lost about both $X^a$ and $X^y$ but not $X^{\varnothing,m}$. Yet, M-bias is rather a theoretic concept and rarely appears in real-world studies (Ding & Miratrix, 2015); therefore, we further concentrate on the loss of confounder information in representations. As described previously, the information on $X^\Delta$ could be lost due to an incorrectly chosen dimensionality of the representation or due to too large balancing with an empirical probability metric. In this case, in addition to the loss of heterogeneity, we have *representation-induced confounding bias* (RICB). That is, the CATE wrt. representations is *non-identifiable* from observational data, $\mathbb{P}(\Phi(X), A, Y)$. This follows from

$$\tau^\phi(\phi) \stackrel{(ii)}{=} \mathbb{E}(Y[1] - Y[0] \mid \phi) \neq \mathbb{E}(Y[1] \mid A = 1, \phi) - \mathbb{E}(Y[0] \mid A = 0, \phi) = \mu_1^\phi(\phi) - \mu_0^\phi(\phi). \quad (9)$$

Technically, both the CATE wrt. representations, $\tau^\phi(\phi)$, and the RICB, $\tau^\phi(\phi) - (\mu_1^\phi(\phi) - \mu_0^\phi(\phi))$, are still identifiable from $\mathbb{P}(X, A, Y)$. Yet, the identification formula has intractable integration in Eq. (8) and the original CATE wrt. covariates, which makes the whole inference senseless.

Motivated by this, we shift our focus in the following section to *partial identification* of the CATE wrt. to representations (or, equivalently, the RICB), as partial identification turns out to be tractable. As a result, we can provide bounds for both quantities. With a slight abuse of formulations, we use 'the bounds on the RICB' and 'the bounds on the representation CATE' interchangeably, as one can be inferred from the other.

**Takeaways.** (1) The minimal sufficient and valid representation would aim to remove only the information about noise and instruments (Ding et al., 2017; Johansson et al., 2022). In low-sample settings, we cannot guarantee that any information is preserved in a low-dimensional representation, so we apriori assume that some information is lost about the sub-covariates. (2) The loss of heterogeneity does not introduce bias but can only make CATE less individualized, namely, suitable only for subgroups. For many applications, like medicine and policy making, subgroup-level CATE is sufficient. (3) The RICB automatically implies a loss of heterogeneity. Therefore, we consider the RICB to be the main problem in representation learning methods for CATE, and, in this paper, we thus aim at providing bounds on the RICB, as the exact value is intractable.

## 4    PARTIAL IDENTIFICATION OF CATE UNDER RICB

In the following, we use the marginal sensitivity model to derive bounds on the RICB (Sec. 4.1). Then, we present a neural refutation framework for estimating the bounds, which can be used with any representation learning method for CATE (Sec. 4.2). Here, we do not assume the specific partitioning of $X$ (which we did before in Sec. 3.1 only for the purpose of providing a better intuition).

### 4.1    BOUNDS ON THE RICB

We now aim to derive lower and upper bounds on the RICB given by $\underline{\tau^\phi}(\phi)$ and $\overline{\tau^\phi}(\phi)$, respectively.[4] For this, we adopt the marginal sensitivity model (MSM) for CATE with binary treatment (Kallus et al., 2019; Jesson et al., 2021; Dorn & Guo, 2022; Oprescu et al., 2023; Frauen et al., 2023).

The MSM assumes that the odds ratio between the covariate (complete) propensity scores and the representation (nominal) propensity scores can be bounded. Applied to our setting, the MSM assumes

$$\Gamma(\phi)^{-1} \le \left(\pi_0^\phi(\phi)/\pi_1^\phi(\phi)\right)\left(\pi_1^x(x)/\pi_0^x(x)\right) \le \Gamma(\phi) \quad \text{for all } x \in \mathcal{X} \text{ s.t. } \Phi(x) = \phi, \quad (10)$$

where $\Gamma(\phi) \ge 1$ is a representation-dependent sensitivity parameter. In our setting, there are no undeserved confounders, and, therefore, the sensitivity parameters can be directly estimated from combined data $\mathbb{P}(X, \Phi(X), A, Y)$.[5]

We make the following observations. For $\Gamma(\phi) = 1$ for all $\phi$, then (1) all the information about the propensity score $\pi_a^x(x)$ is preserved in the representation $\Phi(x)$, and (2) the representation does not contain hidden confounding. If $\Gamma(\phi) \gg 1$, we lose the treatment assignment information and

---

[4]The interval $[\underline{\tau^\phi}(\phi), \overline{\tau^\phi}(\phi)]$ contains intractable $\tau^\phi(\phi)$ and can be inferred from $\mathbb{P}(\Phi(X), A, Y)$ and tractable sensitivity parameters.

[5]Note that this is a crucial *difference* from other applications of the MSM aimed at settings with unobserved confounding, where, instead, the sensitivity parameter is assumed chosen from background knowledge.

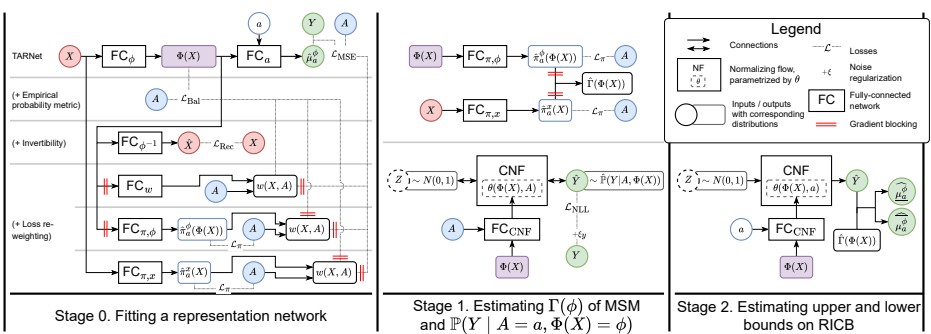

Figure 2: Our neural refutation framework for estimating bounds on the RICB. In **Stage 0**, we fit some representation learning method for CATE, possibly with different constraints like balancing with an empirical probability metric and invertibility, or loss re-weighting. In **Stage 1**, we estimate the sensitivity parameters of the MSM, $\Gamma(\phi)$, and the representation-conditional outcome distribution, $\mathbb{P}(Y \mid A = a, \Phi(x) = \phi)$. In **Stage 2**, we compute the lower and upper bounds on the RICB.

confounding bias could be introduced. Therefore, $\Gamma(\phi)$ indicates how much information is lost about sub-covariates $X^a$ or about $X^\Delta$. In order to actually distinguish whether we lose information specifically about $X^a$ or $X^\Delta$, we would need to know $\mu_a^x(x)$, which makes the task of representation learning for CATE obsolete. Thus, our bounds are conservative in the sense, that they grow if the information on $X^a$ is lost, even though it does not lead to the RICB.

Under the assumption in Eq. (10), bounds (Frauen et al., 2023; Oprescu et al., 2023) on the RICB are given by

$$\underline{\tau^\phi}(\phi) = \underline{\mu_1^\phi}(\phi) - \overline{\mu_0^\phi}(\phi) \qquad \text{and} \qquad \overline{\tau^\phi}(\phi) = \overline{\mu_1^\phi}(\phi) - \underline{\mu_0^\phi}(\phi) \tag{11}$$

with

$$\underline{\mu_a^\phi}(\phi) = \frac{1}{s_-(a,\phi)} \int_{-\infty}^{\mathbb{F}^{-1}(c_- \mid a,\phi)} y\, \mathbb{P}(Y = y \mid a, \phi)\, \mathrm{d}y + \frac{1}{s_+(a,\phi)} \int_{\mathbb{F}^{-1}(c_- \mid a,\phi)}^{+\infty} y\, \mathbb{P}(Y = y \mid a, \phi)\, \mathrm{d}y,$$

$$\overline{\mu_a^\phi}(\phi) = \frac{1}{s_+(a,\phi)} \int_{-\infty}^{\mathbb{F}^{-1}(c_+ \mid a,\phi)} y\, \mathbb{P}(Y = y \mid a, \phi)\, \mathrm{d}y + \frac{1}{s_-(a,\phi)} \int_{\mathbb{F}^{-1}(c_+ \mid a,\phi)}^{+\infty} y\, \mathbb{P}(Y = y \mid a, \phi)\, \mathrm{d}y,$$

where $s_-(a,\phi) = ((1 - \Gamma(\phi))\pi_a^\phi(\phi) + \Gamma(\phi))^{-1}$, $s_+(a,\phi) = ((1 - \Gamma(\phi)^{-1})\pi_a^\phi(\phi) + \Gamma(\phi)^{-1})^{-1}$, $c_- = 1/(1 + \Gamma(\phi))$, $c_+ = \Gamma(\phi)/(1 + \Gamma(\phi))$, $\mathbb{P}(Y = y \mid a, \phi) = \mathbb{P}(Y = y \mid A = a, \Phi(X) = \phi)$ is a representation-conditional density of the outcome, and $\mathbb{F}^{-1}(c \mid a, \phi)$ its corresponding quantile function. This result is an adaptation of the theoretic results, provided in (Frauen et al., 2023; Oprescu et al., 2023). We provide more details on the derivation of the bounds in Lemma 3 of Appendix B. Importantly, the bounds provided Eq. (11) are **valid** and **sharp** (see Corollary 1 in Appendix B). Validity means that our bounds always contain the ground-truth CATE wrt. representations, and sharpness means that the bounds only include CATE wrt. representations induced by the observational distributions complying with the sensitivity constraint from Eq. (10).

The above bounds are not easy to compute (which motivates our neural refutation framework in the following section). The reason is that the bounds on the RICB require inference of so-called conditional values at risk (CVaR) (Artzner et al., 1999; Kallus, 2023). Formally, we have two CVaRs given by $\int_{-\infty}^{q} \mathbb{P}(Y = y \mid a, \phi)$ and $\int_{q}^{+\infty} \mathbb{P}(Y = y \mid a, \phi)$ for $q \in \{\mathbb{F}^{-1}(c_- \mid a, \phi), \mathbb{F}^{-1}(c_+ \mid a, \phi)\}$. CVaRs can be estimated directly without estimating the conditional density, like in (Oprescu et al., 2023). Yet, in our case $c_-$ and $c_+$ depend on $\phi$, and, thus, it is more practical to estimate the conditional density, as done in (Frauen et al., 2023). Given some conditional density estimator, $\hat{\mathbb{P}}(Y = y \mid a, \phi)$, we can then estimate CVaRs via importance sampling, i. e.,

$$\widehat{\text{CVaR}} = \begin{cases} \frac{1}{k} \sum_{i=1}^{\lfloor kc \rfloor} \tilde{y}_i, & \text{for CVaR} = \int_{-\infty}^{\mathbb{F}^{-1}(c \mid a, \phi)} \mathbb{P}(Y = y \mid a, \phi), \\ \frac{1}{k} \sum_{i=\lfloor kc \rfloor + 1}^{k} \tilde{y}_i, & \text{for CVaR} = \int_{\mathbb{F}^{-1}(c \mid a, \phi)}^{+\infty} \mathbb{P}(Y = y \mid a, \phi), \end{cases} \tag{12}$$

where $\{\tilde{y}_i\}_{i=1}^{k}$ is a sorted sample from $\hat{\mathbb{P}}(Y \mid a, \phi)$. Hence, we use a conditional density estimator in our refutation framework.

## 4.2 Neural Refutation Framework for Estimating Bounds

In the following, we provide a flexible neural refutation framework for estimating the bounds on the RICB (see Fig. 2 for the overview). Overall, our refutation framework proceeds in three stages.

Table 2: Results for synthetic experiments. Reported: out-sample policy error rates (with improvements of our bounds); mean over 10 runs. Here, $n_{\text{train}} = 1,000$ and $\delta = 0.0005$.

| | ER$_{\text{out}}$ ($\Delta$ ER$_{\text{out}}$) | |
|---|---|---|
| $d_\phi$ | 1 | 2 |
| TARNet | 30.79% ($-12.89\%$) | 9.82% ($-3.73\%$) |
| BNN (MMD; $\alpha = 0.1$) | 34.32% ($-15.41\%$) | 16.15% ($-4.19\%$) |
| CFR (MMD; $\alpha = 0.1$) | 35.01% ($-14.27\%$) | 11.92% ($-5.54\%$) |
| CFR (MMD; $\alpha = 0.5$) | 35.79% ($-11.43\%$) | 17.89% ($-7.27\%$) |
| CFR (WM; $\alpha = 1.0$) | 34.97% ($-14.27\%$) | 10.88% ($-7.97\%$) |
| CFR (WM; $\alpha = 2.0$) | 35.18% ($-13.63\%$) | 13.19% ($-6.28\%$) |
| InvTARNet | 29.51% ($-0.95\%$) | 5.64% ($-0.02\%$) |
| RCFR (WM; $\alpha = 1.0$) | 33.02% ($-3.58\%$) | 8.00% ($-4.27\%$) |
| CFR-ISW (WM; $\alpha = 1.0$) | 35.00% ($-9.43\%$) | 7.27% ($-1.86\%$) |
| BWCFR (WM; $\alpha = 1.0$) | 34.97% ($-10.02\%$) | 7.44% ($-4.57\%$) |

Classical CATE estimators: $k$-NN: 8.18%; BART: 17.37%; C-Forest: 16.10%

Table 3: Results for HC-MNIST experiments. Reported: out-sample policy error rates (with improvements of our bounds); mean over 10 runs. Here, $\delta = 0.0005$.

| | ER$_{\text{out}}$ ($\Delta$ ER$_{\text{out}}$) | | |
|---|---|---|---|
| $d_\phi$ | 7 | 39 | 78 |
| TARNet | 11.21% ($-2.59\%$) | 10.91% ($-3.34\%$) | 11.01% ($-2.62\%$) |
| BNN (MMD; $\alpha = 0.1$) | 12.00% ($-4.50\%$) | 11.37% ($-5.29\%$) | 20.78% ($-2.01\%$) |
| CFR (MMD; $\alpha = 0.1$) | 11.40% ($-1.89\%$) | 11.05% ($-3.13\%$) | 11.73% ($-4.67\%$) |
| CFR (MMD; $\alpha = 0.5$) | 16.01% ($+19.25\%$) | 12.55% ($-4.95\%$) | 12.90% ($-5.25\%$) |
| CFR (WM; $\alpha = 1.0$) | 24.55% ($-10.42\%$) | 27.87% ($-10.18\%$) | 31.19% ($-11.53\%$) |
| CFR (WM; $\alpha = 2.0$) | 31.71% ($-10.34\%$) | 30.77% ($-7.22\%$) | 31.83% ($-11.91\%$) |
| InvTARNet | 12.18% ($-1.29\%$) | 11.38% ($-3.98\%$) | 11.55% ($-4.34\%$) |
| RCFR (WM; $\alpha = 1.0$) | 21.51% ($-9.17\%$) | 26.97% ($-6.17\%$) | 30.14% ($-14.26\%$) |
| CFR-ISW (WM; $\alpha = 1.0$) | 32.64% ($-10.32\%$) | 26.66% ($-11.30\%$) | 30.02% ($-13.31\%$) |
| BWCFR (WM; $\alpha = 1.0$) | 13.62% ($-3.96\%$) | 28.18% ($+0.24\%$) | 32.54% ($-6.75\%$) |

Lower = better. Improvement over the baseline in green, worsening of the baseline in red
Classical CATE estimators: $k$-NN: 22.34%; BART: 17.51%; C-Forest: 17.65%

**Stage 0.** The initial stage is a naïve application of existing representation learning methods for CATE estimation. As such, we fit a standard representation learning method for the CATE of our choice (e. g., TARNet, CFR, or different variants of CFR). Stage 0 always contains a fully-connected representation subnetwork, FC$_\phi$, which takes covariates $X$ as an input and outputs the representation, $\Phi(X)$. Similarly, potential outcomes predicting subnetwork(s) are also fully-connected, FC$_a$. Together, FC$_\phi$ and FC$_a$ aim to minimize a (potentially weighted) mean squared error of the factual observational data, $\mathcal{L}_{\text{MSE}}$. The representation can be further constrained via (a) balancing with an empirical probability metric, (b) invertibility, or have additional (c) loss re-weighting. These are as follows. (a) Balancing, $\mathcal{L}_{\text{Bal}}$ is implemented with either Wasserstein metric (WM) or maximum mean discrepancy (MMD) with loss coefficient $\alpha$ (Johansson et al., 2016; Shalit et al., 2017). (b) Invertibility (Zhang et al., 2020) is enforced with a reconstruction subnetwork, FC$_{\phi^{-1}}$, and a reconstruction loss, $\mathcal{L}_{\text{Rec}}$. (c) Loss re-weighting is done by employing either trainable weights (Johansson et al., 2018; 2022), with a fully-connected FC$_w$; with a representation-propensity subnetwork (Hassanpour & Greiner, 2019a;b), FC$_{\pi,\phi}$; or with a covariate-propensity subnetwork (Assaad et al., 2021), namely FC$_{\pi,x}$.

**Stage 1.** Here, we use the trained representation subnetwork and then estimate the sensitivity parameters, $\Gamma(\phi)$, and representation-conditional outcome distribution, $\mathbb{P}(Y \mid A = a, \Phi(X) = \phi)$. For that, we train two fully-connected propensity networks, FC$_{\pi,\phi}$ and FC$_{\pi,x}$ (or take them from the Stage 0, if they were used for (c) loss re-weighting). Both networks optimize a binary cross-entropy (BCE) loss, $\mathcal{L}_\pi$. Then, we use Eq. (10) to compute $\hat{\Gamma}(\phi_i)$ for $\phi_i = \Phi(x_i)$ with $x_i$ from $\mathcal{D}$. Specifically, each $\hat{\Gamma}(\phi_i)$ is a maximum over all $\hat{\Gamma}(\Phi(x_j))$, where $\Phi(x_j)$ are the representations of the training sample in $\delta$-ball around $\phi_i$. Here, $\delta$ is a hyperparameter, whose misspecification only makes the bounds more conservative but does not influence their validity. In addition, we estimate the representation-conditional outcome density with a conditional normalizing flow (CNF) (Trippe & Turner, 2018) using a fully-connected context subnetwork, FC$_{\text{CNF}}$. The latter aims at minimizing the negative log-likelihood of the observational data, $\mathcal{L}_{\text{NLL}}$.

**Stage 2.** Finally, we compute the lower and upper bounds on the RICB, as described in Eq.(11)–(12). Here, the CNF is beneficial for our task, as it enables direct sampling from the estimated conditional distribution. Further details on implementation details and hyperparameter tuning are in Appendix C.

## 5 EXPERIMENTS

**Setup.** We empirically validate our bounds on the RICB.[6] For that, we use several (semi-)synthetic benchmarks with ground-truth counterfactual outcomes $Y[0]$ and $Y[1]$ and ground-truth CATE $\tau^x(x)$. Inspired by Jesson et al. (2021), we designed our experiments so that we compare policies based on (a) the estimated CATE or (b) the estimated bounds on the RICB. In (a), a policy based on the point estimate of the CATE applies a treatment whenever the CATE is positive, i.e. $\hat{\pi}(\phi) = \mathbb{1}\{\widehat{\tau^\phi}(\phi) > 0\}$. In (b), a policy $\pi^*(\phi)$ based on the bounds on the RICB has three decisions: (1) to treat, i.e., when $\widehat{\underline{\tau^\phi}}(\phi) > 0$; (2) to do nothing, i.e., when $\widehat{\overline{\tau^\phi}}(\phi) < 0$; and (3) to defer a decision, otherwise.

---

[6]We also considered other evaluation techniques. However, the ground-truth CATE wrt. representations is intractable; therefore, one can *not* employ established metrics such as coverage or compare MSEs directly. As a remedy, we follow Jesson et al. (2021) and evaluate the decision-making under our CATE, which is closely aligned with how our refutation framework would be used for making reliable decisions in practice.

Table 4: Results for IHDP100 experiments. Reported: out-sample policy error rates (with improvements of our bounds); mean over 100 train/test splits. Here, $\delta = 0.0005$.

| | $\mathrm{ER_{out}}$ $(\Delta\ \mathrm{ER_{out}})$ | | | | |
|---|---|---|---|---|---|
| $d_\phi$ | 5 | 10 | 15 | 20 | 25 |
| TARNet | 3.17% (−2.65%) | 2.88% (−2.30%) | 3.28% (−2.74%) | 3.23% (−2.52%) | 2.89% (−2.37%) |
| BNN (MMD; $\alpha = 0.1$) | 2.32% (−1.49%) | 2.43% (−1.40%) | 2.59% (−2.03%) | 2.43% (−1.87%) | 2.29% (−1.16%) |
| CFR (MMD; $\alpha = 0.1$) | 1.77% (−0.89%) | 2.09% (−1.03%) | 2.23% (−1.63%) | 1.88% (−0.48%) | 2.04% (−1.46%) |
| CFR (MMD; $\alpha = 0.5$) | 2.07% (−1.46%) | 2.00% (+3.98%) | 2.68% (+1.89%) | 2.36% (+6.37%) | 2.17% (+3.41%) |
| CFR (WM; $\alpha = 1.0$) | 1.93% (−0.89%) | 1.75% (−0.25%) | 1.83% (−1.24%) | 1.83% (−0.49%) | 1.80% (−0.20%) |
| CFR (WM; $\alpha = 2.0$) | 1.97% (−0.04%) | 2.17% (−1.49%) | 2.05% (−1.21%) | 2.08% (−1.29%) | 2.09% (−1.36%) |
| InvTARNet | 2.52% (−1.95%) | 3.11% (−2.47%) | 2.99% (−2.51%) | 2.79% (−2.41%) | 2.83% (−2.28%) |
| RCFR (WM; $\alpha = 1.0$) | 3.36% (−2.84%) | 3.45% (−1.52%) | 2.67% (−1.57%) | 4.69% (−3.83%) | 1.95% (+1.06%) |
| CFR-ISW (WM; $\alpha = 1.0$) | 2.24% (−0.96%) | 1.93% (−0.68%) | 1.71% (−1.18%) | 1.85% (−1.54%) | 1.88% (−0.19%) |
| BWCFR (WM; $\alpha = 1.0$) | 3.57% (−1.49%) | 3.52% (−2.16%) | 3.88% (−1.10%) | 3.80% (−2.38%) | 4.07% (−1.18%) |

Lower = better. Improvement over the baseline in green, worsening of the baseline in red
Classical CATE estimators: $k$-NN: 7.47%; BART: 5.07%; C-Forest: 6.28%

**Evaluation metric.** To compare our bounds with the point estimates, we employ an error rate of the policy (ER). ER is defined as the rate of how often decisions of the estimated policy are different from the decision of the optimal policy, $\pi(x) = \mathbb{1}\{\tau^x(x) > 0\}$. For the policy based on our bounds, we report the error rate on the non-deferred decisions. Hence, improvements over the baselines due to our refutation framework would imply that our bounds are precise and that we defer the right observations/individuals. Additionally, we report a root precision in estimating treatment effect (rPEHE) of original methods in Appendix E.

**Baselines.** We combine our refutation framework with the state-of-the-art baselines from representation learning. These are: **TARNet** (Shalit et al., 2017; Johansson et al., 2022) implements a representation network without constraints; **BNN** (Johansson et al., 2016) enforces balancing with MMD ($\alpha = 0.1$); **CFR** (Shalit et al., 2017; Johansson et al., 2022) is used in four variants of balancing with WM ($\alpha = 1.0/2.0$) and with MMD ($\alpha = 0.1/0.5$). **InvTARNet** adds an invertibility constraint to the TARNet via the reconstruction loss, as in (Zhang et al., 2020). Finally, three methods use additional loss re-weighting on top of the balancing with WM ($\alpha = 1.0$): **RCFR** (Johansson et al., 2018; 2022), **CFR-ISW** (Hassanpour & Greiner, 2019a), and **BWCFR** (Assaad et al., 2021). Also, for reference, we provide results for classical (non-neural) CATE estimators, i. e., $k$**-NN** regression, Bayesian additive regression trees (**BART**) (Chipman et al., 2010), and causal forests (**C-Forest**) (Wager & Athey, 2018).

■ **Synthetic data.** We adapt the synthetic data generator from Kallus et al. (2019), where we add an unobserved confounder to the observed covariates so that $d_x = 2$. We sample $n_{train} \in \{500; 1,000; 5,000; 10,000\}$ training and $n_{test} = 1,000$ datapoints. Further details are in Appendix D. We plot ground-truth and estimated decision boundaries in Appendix E. Additionally, in Table 2, we provide error rates of the original representation learning methods and improvements in error rates with our bounds. Our refutation framework achieves clear improvements in the error rate among all the baselines. This improvement is especially large, when $d_\phi = 1$, so that there are both the loss of heterogeneity and the RICB.

■ **IHDP100 dataset.** The Infant Health and Development Program (IHDP) (Hill, 2011; Shalit et al., 2017) is a classical benchmark for CATE estimation and consists of the 100 train/test splits with $n_{train} = 672, n_{test} = 75, d_x = 25$. Again, our refutation framework improves the error rates for almost all of the baselines (Table 4). We observe one deviation for CFR (MMD; $\alpha = 0.5$), but this can be explained by too large balancing with an empirical probability metric and the low-sample size.

■ **HC-MNIST dataset.** HC-MNIST is a semi-synthetic benchmark on top of the MNIST image dataset (Jesson et al., 2021). We consider all covariates as observed (see Appendix D). The challenge in CATE estimation comes from the high-dimensionality of covariates, i. e., $d_x = 784 + 1$. Again, our refutation framework improves over the baselines by a clear margin (see Table 3).

**Additional results.** In Appendix E, we provide additional results, where we report deferral rates in addition to the error rates for different values of the hyperparameter $\delta$. Our refutation framework reduces the error rates while increasing the deferral rates not too much, which demonstrates its effectiveness.

**Conclusion.** We studied the validity of representation learning for CATE estimation. The validity may be violated due to low-dimensional representations as these introduce a *representation-induced confounding bias*. As a remedy, we introduced a novel, representation-agnostic refutation framework that practitioners can use to estimate bounds on the RICB and thus improve the reliability of their CATE estimator.

**Acknowledgments.** S.F. acknowledges funding via Swiss National Science Foundation Grant 186932.

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

## A    EXTENDED RELATED WORK

**CATE estimation.** A wide range of machine learning methods have been employed for estimating CATE. Examples include tailored methods built on top of linear regression Johansson et al. (2016), regression trees (Chipman et al., 2010; Hill, 2011), and random forests (Wager & Athey, 2018). Alaa & van der Schaar (2017) proposed non-parametric approaches based on Gaussian processes. Later, it was shown in (Alaa & van der Schaar, 2018a;b), that Gaussian processes can achieve optimal minimax rates for non-parametric estimation, but only in asymptotic regime and under smoothness conditions.[7] Another stream of literature studied meta-learners (Künzel et al., 2019; Kennedy, 2023; Nie & Wager, 2021; Kennedy et al., 2022), which combine different nuisance function estimators; and double/debiased machine learning (Chernozhukov et al., 2017), which turns CATE estimation into two-stage regression. Both approaches can achieve optimal convergence rates of CATE estimation, but, again, only asymptotically and with well-specified models. Curth & van der Schaar (2021a) and Curth & van der Schaar (2021b) specifically studied neural networks as base model classes for meta-learners. According to (Curth & van der Schaar, 2021b), representation learning methods are one-step (plug-in) learners and they aim at best-in-class estimation of CATE, given potentially misspecified parametric model classes (Johansson et al., 2022).

**Prognostic scores.** Inferring prognostic scores (Hansen, 2008) is relevant to CATE estimation with representation learning. The prognostic score is such a transformation of covariates, which preserves all the information about a potential outcome and can be seen as a sufficient dimensionality reduction technique for CATE. For example, linear prognostic scores were rigorously studied in (Hu et al., 2014; Huang & Chan, 2017). As mentioned by (Johansson et al., 2022), representation learning CATE estimators aim to learn approximate non-linear prognostic scores.

**Uncertainty of CATE estimation.** Several works studied uncertainty around CATE estimation. For example, Jesson et al. (2020) studied epistemic uncertainty, and Jesson et al. (2021) additionally considered bias due to unobserved confounding, which was a part of aleatoric uncertainty. Our work differs from both of them, as we aim at estimating the confounding bias, present in low-dimensional (potentially constrained) representations. This bias still persists in an infinite-sample regime, where epistemic uncertainty drops to zero.

**Runtime confounding.** Runtime confounding (Coston et al., 2020) arises when a subset of covariates is not available during prediction time. This setting is highly relevant, e. g., for multi-step-ahead prediction of time-varying methods where future time-varying covariates are not observed; and (questionably) some methods used balanced representations as a heuristic to circumvent this bias (Bica et al., 2020; Melnychuk et al., 2022; Seedat et al., 2022; Hess et al., 2024). Notably, non-availability of covariates can be seen as a special case of low-dimensional representations which zero-out some inputs and have identity transformations for the others. Therefore, runtime confounding is a special case of our proposed RICB.

---

[7]One important result of (Alaa & van der Schaar, 2018b) is that minimax rate of non-parametric estimation of CATE does not asymptotically depend on *treatment imbalance*. In a low-sample regime nevertheless, addressing treatment imbalance is still important.

## B  PROOFS

**Lemma 1.** *Let $X$ be partitioned in a cluster of sub-covariates, $X = \{X^\varnothing, X^a, X^y, X^\Delta\}$. If the following independencies hold*

$$X \perp\!\!\!\perp X^\varnothing \mid \Phi(X), \quad X \perp\!\!\!\perp X^a \mid \Phi(X), \quad X \perp\!\!\!\perp X^y \mid \Phi(X), \quad X \perp\!\!\!\perp X^\Delta \mid \Phi(X), \tag{13}$$

*then $\Phi(\cdot)$ is an invertible transformation.*

*Proof.* Without a loss of generality, let us consider a posterior distribution of some sub-covariate, e. g., $X^a$, given a representation $\Phi(X) = \phi$, i. e., $\mathbb{P}(X^a \mid \Phi(X) = \phi)$. Due to the independence, the following holds

$$\mathbb{P}(X^a \mid \Phi(X) = \phi) = \mathbb{P}(X^a \mid \Phi(X) = \phi, X = x) = \mathbb{P}(X^a \mid X = x), \tag{14}$$

where the last equality follows from the fact that $\Phi(\cdot)$ is a measurable function. It is easy to see that $\mathbb{P}(X^a \mid X = x)$ is a Dirac-delta distribution. Therefore, for every $\phi$, $\mathbb{P}(X^a \mid \Phi(X) = \phi)$ is also a Dirac-delta distribution, namely it puts a point mass on some $x^a$. In this way, we can define an inverse of $\Phi(\cdot)$ wrt. $\phi$.

The equalities in Eq. (14) also hold for other sub-covariates of $X$, and, thus, we can construct a full inverse of $\Phi(\cdot)$. $\qquad\square$

**Lemma 2** (Removal of noise and instruments). *Let $X = \{X^\varnothing, X^a, X^y, X^\Delta\}$ be partitioned according to the clustered causal diagram in Fig. 1. Then, representation $\Phi(\cdot)$ is valid if*

$$X \perp\!\!\!\perp X^\Delta \mid \Phi(X), \quad X \perp\!\!\!\perp X^y \mid \Phi(X). \tag{15}$$

*Proof.* First, it is easy to see that the potential outcomes are d-separated from $X$ by conditioning on $X^\Delta$ and $X^y$, i. e., $X \perp\!\!\!\perp Y[a] \mid X^\Delta, X^y$ or, equivalently, $\mathbb{P}(Y[a] \mid X = x) = \mathbb{P}(Y[a] \mid X^\Delta = x^\Delta, X^y = x^y)$. This is an immediate result of applying a d-separation criterion to an extended causal diagram in Fig. 3 (also known as parallel worlds network), i. e., $X^\Delta$ and $X^y$ d-separate $Y[a]$ from $X^\varnothing$ and $X^a$. Therefore, the following holds

$$\tau^x(x) = \mathbb{E}(Y[1] - Y[0] \mid X = x) = \mathbb{E}(Y[1] - Y[0] \mid X^\Delta = x^\Delta, X^y = x^y). \tag{16}$$

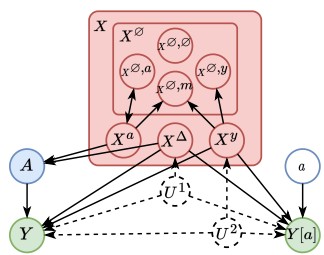

Figure 3: Parallel worlds network with observed outcome $Y$ and potential outcome $Y[a]$. In the diagram, (1) $X^\Delta$ and $X^y$ d-separate $Y[a]$ from $X^\varnothing$ and $X^a$; and (2) $X^\Delta$ and $X^y$ d-separate $Y[a]$ from $A$.

Furthermore, due to the invertibility of $\Phi(\cdot)$ wrt. $X^\Delta$ and $X^y$, which follows from Eq. (15) and Lemma 1, we have

$$\mathbb{E}(Y[1] - Y[0] \mid X^\Delta = x^\Delta, X^y = x^y) = \mathbb{E}(Y[1] - Y[0] \mid \Phi(X) = \Phi(x)) = \tau^\phi(\Phi(x)). \tag{17}$$

Second, $Y[a]$ is also d-separated from $A$ by $X^\Delta$ and $X^y$ (see Fig. 3). Therefore, we can employ assumptions (1)–(3) and substitute potential outcomes with observed outcomes:

$$\mathbb{E}(Y[1] - Y[0] \mid X^\Delta = x^\Delta, X^y = x^y) \tag{18}$$

$$= \mathbb{E}(Y \mid A = 1, X^\Delta = x^\Delta, X^y = x^y) - \mathbb{E}(Y \mid A = 0, X^\Delta = x^\Delta, X^y = x^y) \tag{19}$$

$$= \mu_1^\phi(\Phi(x)) - \mu_0^\phi(\Phi(x)), \tag{20}$$

where the latter holds due to the invertibility. $\qquad\square$

**Lemma 3** (MSM bounds on the RICB). *Let a representation $\phi = \Phi(x)$ satisfy the following sensitivity constraint:*

$$\Gamma(\phi)^{-1} \leq \left(\pi_0^\phi(\phi)/\pi_1^\phi(\phi)\right)\left(\pi_1^x(x)/\pi_0^x(x)\right) \leq \Gamma(\phi) \quad \text{for all } x \in \mathcal{X} \text{ s.t. } \Phi(x) = \phi, \quad (21)$$

*where $\Gamma(\phi) \geq 1$ is a representation-dependent sensitivity parameter.*

*Under the assumption in Eq.* (21)*, bounds (Frauen et al., 2023; Oprescu et al., 2023) on the RICB are given by*

$$\underline{\tau}^\phi(\phi) = \underline{\mu_1^\phi}(\phi) - \overline{\mu_0^\phi}(\phi) \qquad and \qquad \overline{\tau}^\phi(\phi) = \overline{\mu_1^\phi}(\phi) - \underline{\mu_0^\phi}(\phi) \quad (22)$$

*with*

$$\underline{\mu_a^\phi}(\phi) = \frac{1}{s_-(a,\phi)} \int_{-\infty}^{\mathbb{F}^{-1}(c_-\,|a,\phi)} y\,\mathbb{P}(Y = y \mid a, \phi)\,\mathrm{d}y + \frac{1}{s_+(a,\phi)} \int_{\mathbb{F}^{-1}(c_-|a,\phi)}^{+\infty} y\,\mathbb{P}(Y = y \mid a, \phi)\,\mathrm{d}y,$$

$$\overline{\mu_a^\phi}(\phi) = \frac{1}{s_+(a,\phi)} \int_{-\infty}^{\mathbb{F}^{-1}(c_+\,|a,\phi)} y\,\mathbb{P}(Y = y \mid a, \phi)\,\mathrm{d}y + \frac{1}{s_-(a,\phi)} \int_{\mathbb{F}^{-1}(c_+|a,\phi)}^{+\infty} y\,\mathbb{P}(Y = y \mid a, \phi)\,\mathrm{d}y,$$

*where $s_-(a,\phi) = ((1 - \Gamma(\phi))\pi_a^\phi(\phi) + \Gamma(\phi))^{-1}$, $s_+(a,\phi) = ((1 - \Gamma(\phi)^{-1})\pi_a^\phi(\phi) + \Gamma(\phi)^{-1})^{-1}$, $c_- = 1/(1 + \Gamma(\phi))$, $c_+ = \Gamma(\phi)/(1 + \Gamma(\phi))$, $\mathbb{P}(Y = y \mid a, \phi) = \mathbb{P}(Y = y \mid A = a, \Phi(X) = \phi)$ is a representation-conditional density of the outcome, and $\mathbb{F}^{-1}(c \mid a, \phi)$ its corresponding quantile function.*

*Proof.* We employ the main results from (Frauen et al., 2023).

First, it is easy to see, that our definition of the MSM, i.e., Eq. (21), fits to the definition of the generalized marginal sensitivity model (GMSM) from (Frauen et al., 2023). Specifically, we can rearrange the terms in Eq. (21) in the following way (see Appendix C.1 in (Frauen et al., 2023)):

$$\frac{1}{(1 - \Gamma)\,\pi_a^\phi(\phi) + \Gamma} \leq \frac{\pi_a^x(x)}{\pi_a^\phi(\phi)} \leq \frac{1}{(1 - \Gamma^{-1})\,\pi_a^\phi(\phi) + \Gamma^{-1}} \quad \text{for all } x \in \mathcal{X} \text{ s.t. } \Phi(x) = \phi, \quad (23)$$

where $a \in \{0, 1\}$. Then, we apply the Bayes rule to the inner ratio

$$\frac{\pi_a^x(x)}{\pi_a^\phi(\phi)} = \frac{\mathbb{P}(A = a \mid X = x, \Phi(X) = \Phi(x))}{\mathbb{P}(A = a \mid \Phi(X) = \Phi(x))} \quad (24)$$

$$= \frac{\mathbb{P}(X = x \mid \Phi(X) = \Phi(x), A = a)\,\pi_a^\phi(\phi)}{\mathbb{P}(X = x \mid \Phi(X) = \Phi(x))\,\pi_a^\phi(\phi)} \quad (25)$$

$$= \frac{\mathbb{P}(X = x \mid \Phi(X) = \Phi(x), A = a)}{\mathbb{P}(X = x \mid \Phi(X) = \Phi(x), do(A = a))}, \quad (26)$$

where the latest equality holds, as the intervention on $A$, $do(A = a)$, does not change the conditional distribution of $\mathbb{P}(X \mid \Phi(X) = \Phi(x))$. Therefore, our definition of the MSM matches the definitions of the GMSM in (Frauen et al., 2023).

Second, to derive the bound on the RICB, we refer to Theorem 1 in (Frauen et al., 2023), which states that the lower and upper bounds are expectations of maximally left- and right-shifted interventional distributions, respectively. The maximally left-shifted distribution, $\mathbb{P}_-(Y \mid a, \phi)$, and the maximally right-shifted distribution, $\mathbb{P}_+(Y \mid a, \phi)$, are defined in the following way:

$$\mathbb{P}_-(Y = y \mid a, \phi) = \begin{cases} (1/s_-(a,\phi))\,\mathbb{P}(Y = y \mid a, \phi), & \text{if } \mathbb{F}(y \mid a, \phi) \leq c_-, \\ (1/s_+(a,\phi))\,\mathbb{P}(Y = y \mid a, \phi), & \text{if } \mathbb{F}(y \mid a, \phi) > c_-, \end{cases} \quad (27)$$

$$\mathbb{P}_+(Y = y \mid a, \phi) = \begin{cases} (1/s_+(a,\phi))\,\mathbb{P}(Y = y \mid a, \phi), & \text{if } \mathbb{F}(y \mid a, \phi) \leq c_+, \\ (1/s_-(a,\phi))\,\mathbb{P}(Y = y \mid a, \phi), & \text{if } \mathbb{F}(y \mid a, \phi) > c_+, \end{cases} \quad (28)$$

where $s_-(a,\phi) = ((1 - \Gamma(\phi))\pi_a^\phi(\phi) + \Gamma(\phi))^{-1}$, $s_+(a,\phi) = ((1 - \Gamma(\phi)^{-1})\pi_a^\phi(\phi) + \Gamma(\phi)^{-1})^{-1}$, $c_- = 1/(1 + \Gamma(\phi))$, $c_+ = \Gamma(\phi)/(1 + \Gamma(\phi))$, $\mathbb{P}(Y = y \mid a, \phi) = \mathbb{P}(Y = y \mid A = a, \Phi(X) = \phi)$ is a representation-conditional density of the outcome, and $\mathbb{F}(y \mid a, \phi)$ its corresponding CDF. Then, the lower and upper bounds can be obtained by taking expectations, i.e.,

$$\underline{\mu_a^\phi}(\phi) = \int_{\mathcal{Y}} y\,\mathbb{P}_-(Y = y \mid a, \phi)\,\mathrm{d}y \quad \text{and} \quad \overline{\mu_a^\phi}(\phi) = \int_{\mathcal{Y}} y\,\mathbb{P}_+(Y = y \mid a, \phi)\,\mathrm{d}y. \quad (29)$$

$\square$

**Corollary 1** (Validity and sharpness of bounds)**.** *The bounds on the RICB in Lemma 3 are valid and sharp. Validity implies that the bounds always contain the ground-truth CATE wrt. representations, i. e., $\tau^\phi(\phi) \in [\underline{\tau^\phi}(\phi), \overline{\tau^\phi}(\phi)]$. Sharpness implies that all the values in the ignorance interval $[\underline{\tau^\phi}(\phi), \overline{\tau^\phi}(\phi)]$ are induced by the distributions which comply with the sensitivity constraint from Eq. (10).*

*Proof.* We refer to (Frauen et al., 2023) for a formal proof of both properties. □

## C  IMPLEMENTATION AND HYPERPARAMETERS

**Implementation.** We implemented our refutation framework in PyTorch and Pyro. For better compatibility, the fully-connected subnetworks have one hidden layer with a tuneable number of units. For the CNF, we employed neural spline flows (Durkan et al., 2019) with a standard normal as a base distribution and noise regularization (Rothfuss et al., 2019). The number of knots and the intensities of the noise regularization are hyperparameters. The CNF is trained via stochastic gradient descent (SGD) and all the other networks with AdamW (Loshchilov & Hutter, 2019). Each network was trained with $n_{\text{iter}} = 5,000$ train iterations.

**Hyperparameters.** We performed hyperparameter tuning at all the stages of our refutation framework for all the networks based on five-fold cross-validation using the training subset. At each stage, we did a random grid search with respect to different tuning criteria. Table 5 provides all the details on hyperparameters tuning. For reproducibility, we made tuned hyperparameters available in our GitHub.[8]

Table 5: Hyperparameter tuning for baselines and our refutation framework.

| Stage | Model | Hyperparameter | Range / Value |
|---|---|---|---|
| **Stage 0** | TARNet BNN CFR InvTARNet BWCFR | Learning rate | 0.001, 0.005, 0.01 |
| | | Minibatch size | 32, 64, 128 |
| | | Weight decay | 0.0, 0.001, 0.01, 0.1 |
| | | Hidden units in $\text{FC}_\phi$ ($\text{FC}_{\phi^{-1}}$) | $R\,d_x$, 1.5 $Rd_x$, 2 $Rd_x$ |
| | | Hidden units in $\text{FC}_a$ | $R\,d_\phi$, 1.5 $Rd_\phi$, 2 $Rd_\phi$ |
| | | Tuning strategy | random grid search with 50 runs |
| | | Tuning criterion | factual MSE loss |
| | | Optimizer | AdamW |
| | CFR-ISW | Representation network learning rate | 0.001, 0.005, 0.01 |
| | | Propensity network learning rate | 0.001, 0.005, 0.01 |
| | | Minibatch size | 32, 64, 128 |
| | | Representation network weight decay | 0.0, 0.001, 0.01, 0.1 |
| | | Propensity network weight decay | 0.0, 0.001, 0.01, 0.1 |
| | | Hidden units in $\text{FC}_\phi$ | $R\,d_x$, 1.5 $Rd_x$, 2 $Rd_x$ |
| | | Hidden units in $\text{FC}_a$ | $R\,d_\phi$, 1.5 $Rd_\phi$, 2 $Rd_\phi$ |
| | | Hidden units in $\text{FC}_{\pi,\phi}$ | $R\,d_\phi$, 1.5 $Rd_\phi$, 2 $Rd_\phi$ |
| | | Tuning strategy | random grid search with 100 runs |
| | | Tuning criterion | factual MSE loss + factual BCE loss |
| | | Optimizer | AdamW |
| | RCFR | Learning rate | 0.001, 0.005, 0.01 |
| | | Minibatch size | 32, 64, 128 |
| | | Weight decay | 0.0, 0.001, 0.01, 0.1 |
| | | Hidden units in $\text{FC}_\phi$ | $R\,d_x$, 1.5 $Rd_x$, 2 $Rd_x$ |
| | | Hidden units in $\text{FC}_a$ | $R\,d_\phi$, 1.5 $Rd_\phi$, 2 $Rd_\phi$ |
| | | Hidden units in $\text{FC}_w$ | $R\,d_\phi$, 1.5 $Rd_\phi$, 2 $Rd_\phi$ |
| | | Tuning strategy | random grid search with 50 runs |
| | | Tuning criterion | factual MSE loss |
| | | Optimizer | AdamW |
| **Stage 1** | Propensity networks | Learning rate | 0.001, 0.005, 0.01 |
| | | Minibatch size | 32, 64, 128 |
| | | Weight decay | 0.0, 0.001, 0.01, 0.1 |
| | | Hidden units in $\text{FC}_{\pi,\phi|x}$ | $R\,d_{\phi|x}$, 1.5 $Rd_{\phi|x}$, 2 $Rd_{\phi|x}$ |
| | | Tuning strategy | random grid search with 50 runs |
| | | Tuning criterion | factual BCE loss |
| | | Optimizer | AdamW |
| | CNF | Learning rate | 0.001, 0.005, 0.01 |
| | | Minibatch size | 32, 64, 128 |
| | | Hidden units in $\text{FC}_{\text{CNF}}$ | $R\,d_\phi$, 1.5 $Rd_\phi$, 2 $Rd_\phi$ |
| | | Number of knots | 5, 10, 20 |
| | | Intensity of outcome noise | 0.05, 0.1, 0.5 |
| | | Intensity of representation noise | 0.05, 0.1, 0.5 |
| | | Tuning strategy | random grid search with 100 runs |
| | | Tuning criterion | factual negative log-likelihood loss |
| | | Optimizer | SGD (momentum = 0.9) |

$R = 2$ (synthetic data), $R = 1$ (IHDP100, HC-MNIST datasets)

---

[8] https://github.com/Valentyn1997/RICB.

# D DATASET DETAILS

## D.1 SYNTHETIC DATA

We adopt a synthetic benchmark with hidden confounding from (Kallus et al., 2019), but we add the confounder as the second observed covariate. Specifically, synthetic covariates, $X_1, X_2$, a treatment, $A$, and an outcome, $Y$, are sampled from the following data generating mechanism:

$$\begin{cases} X_1 \sim \text{Unif}(-2, 2), \\ X_2 \sim N(0, 1), \\ A \sim \text{Bern}\left(\frac{1}{1+\exp(-(0.75\,X_1-X_2+0.5))}\right) \\ Y \sim N\big((2\,A-1)\,X_1 + A - 2\,\sin(2\,(2\,A-1)\,X_1+X_2) - 2\,X_2\,(1+0.5\,X_1), 1\big), \end{cases} \tag{30}$$

where $X_1, X_2$ are mutually independent.

## D.2 HC-MNIST DATASET

Jesson et al. (2021) introduced a complex high-dimensional, semi-synthetic dataset based on the MNIST image dataset (LeCun, 1998), namely HC-MNIST. The MNIST dataset contains $n_{\text{train}} = 60,000$ train and $n_{\text{test}} = 10,000$ test images. HC-MNIST takes original high-dimensional images and maps them onto a one-dimensional manifold, where potential outcomes depend in a complex way on the average intensity of light and the label of an image. The treatment also uses this one-dimensional summary, $\phi$, together with an additional (hidden) synthetic confounder, $U$ (we consider this hidden confounder as another observed covariate). HC-MNIST is then defined by the following data-generating mechanism:

$$\begin{cases} U \sim \text{Bern}(0.5), \\ X \sim \text{MNIST-image}(\cdot), \\ \phi := \left(\text{clip}\left(\frac{\mu_{N_x}-\mu_c}{\sigma_c}; -1.4, 1.4\right) - \text{Min}_c\right)\frac{\text{Max}_c-\text{Min}_c}{1.4-(-1.4)}, \\ \alpha(\phi; \Gamma^*) := \frac{1}{\Gamma^* \text{sigmoid}(0.75\phi+0.5)} + 1 - \frac{1}{\Gamma^*}, \\ \beta(\phi; \Gamma^*) := \frac{\Gamma^*}{\text{sigmoid}(0.75\phi+0.5)} + 1 - \Gamma^*, \\ A \sim \text{Bern}\left(\frac{u}{\alpha(\phi;\Gamma^*)} + \frac{1-u}{\beta(\phi;\Gamma^*)}\right), \\ Y \sim N\big((2A-1)\phi + (2A-1) - 2\sin(2(2A-1)\phi) - 2(2U-1)(1+0.5\phi), 1\big), \end{cases} \tag{31}$$

where $c$ is a label of the digit from the sampled image $X$; $\mu_{N_x}$ is the average intensity of the sampled image; $\mu_c$ and $\sigma_c$ are the mean and standard deviation of the average intensities of the images with the label $c$; and $\text{Min}_c = -2 + \frac{4}{10}c, \text{Max}_c = -2 + \frac{4}{10}(c+1)$. The parameter $\Gamma^*$ defines what factor influences the treatment assignment to a larger extent, i.e., the additional confounder or the one-dimensional summary. We set $\Gamma^* = \exp(1)$. For further details, we refer to (Jesson et al., 2021).

# E  ADDITIONAL RESULTS

## E.1  PRECISION IN ESTIMATING TREATMENT EFFECT

In the following, we report results on the root precision in estimating treatment effect (rPEHE) for the baseline representation learning CATE estimators. The rPEHE is given by $\sqrt{\frac{1}{n}\sum_{i=1}^{n}\left((y_i[1]-y_i[0])-\widehat{\tau^\phi}(\Phi(x_i))\right)^2}$, where $y_i[0], y_i[1]$ are sample from both potential outcomes. Tables 6, 7, and 8 report the results for synthetic data, IHDP100 dataset, and HC-MNIST dataset, respectively. For comparison, we also provide the performance of oracle estimators, i. e., ground-truth CATE. Notably, there is no universally best method among all of the benchmarks.

Table 6: Results for synthetic experiments. Reported: in-sample and out-sample rPEHE; mean over 10 runs.

| | | $\text{rPEHE}_{\text{in}}$ | | | | | | |
|---|---|---|---|---|---|---|---|---|
| $d_\phi$ | | 1 | | | | 2 | | |
| $n_{\text{train}}$ | 500 | 1000 | 5000 | 10000 | 500 | 1000 | 5000 | 10000 |
| $k$-NN | **0.54 ± 0.03** | **0.51 ± 0.02** | **0.48 ± 0.01** | **0.47 ± 0.01** | **0.54 ± 0.03** | **0.51 ± 0.02** | **0.48 ± 0.01** | **0.47 ± 0.01** |
| BART | 0.64 ± 0.04 | 0.60 ± 0.03 | 0.51 ± 0.01 | 0.48 ± 0.01 | 0.64 ± 0.04 | 0.60 ± 0.03 | 0.51 ± 0.01 | 0.48 ± 0.01 |
| C-Forest | 0.67 ± 0.04 | 0.59 ± 0.03 | 0.49 ± 0.01 | 0.48 ± 0.00 | 0.67 ± 0.04 | 0.59 ± 0.03 | 0.49 ± 0.01 | 0.48 ± 0.00 |
| TARNet | 0.96 ± 0.05 | 0.90 ± 0.07 | 0.90 ± 0.07 | 0.91 ± 0.05 | 0.59 ± 0.10 | 0.58 ± 0.06 | 0.55 ± 0.05 | 0.56 ± 0.04 |
| BNN (MMD; $\alpha = 0.1$) | 0.98 ± 0.05 | 0.98 ± 0.06 | 0.97 ± 0.01 | 0.97 ± 0.01 | 0.70 ± 0.11 | 0.64 ± 0.11 | 0.63 ± 0.08 | 0.59 ± 0.09 |
| CFR (MMD; $\alpha = 0.1$) | 0.96 ± 0.05 | 0.97 ± 0.05 | 0.93 ± 0.05 | 0.95 ± 0.01 | 0.61 ± 0.06 | 0.61 ± 0.08 | 0.61 ± 0.06 | 0.59 ± 0.09 |
| CFR (MMD; $\alpha = 0.5$) | 0.97 ± 0.05 | 0.98 ± 0.06 | 0.97 ± 0.01 | 0.97 ± 0.01 | 0.66 ± 0.07 | 0.72 ± 0.07 | 0.66 ± 0.10 | 0.67 ± 0.07 |
| CFR (WM; $\alpha = 1.0$) | 0.95 ± 0.04 | 0.97 ± 0.06 | 0.95 ± 0.01 | 0.95 ± 0.01 | 0.56 ± 0.05 | 0.57 ± 0.06 | 0.60 ± 0.08 | 0.53 ± 0.03 |
| CFR (WM; $\alpha = 2.0$) | 0.95 ± 0.05 | 0.96 ± 0.05 | 0.97 ± 0.04 | 0.96 ± 0.01 | 0.58 ± 0.06 | 0.61 ± 0.06 | 0.58 ± 0.08 | 0.57 ± 0.07 |
| InvTARNet | 0.90 ± 0.05 | 0.90 ± 0.07 | 0.87 ± 0.05 | 0.88 ± 0.06 | **0.54 ± 0.05** | **0.51 ± 0.03** | 0.51 ± 0.04 | 0.49 ± 0.01 |
| RCFR (WM; $\alpha = 1.0$) | 0.97 ± 0.06 | 0.97 ± 0.05 | 0.96 ± 0.01 | 0.95 ± 0.03 | 0.57 ± 0.05 | 0.55 ± 0.03 | 0.55 ± 0.03 | 0.55 ± 0.05 |
| CFR-ISW (WM; $\alpha = 1.0$) | 0.96 ± 0.04 | 0.97 ± 0.05 | 0.99 ± 0.10 | 0.96 ± 0.01 | **0.54 ± 0.03** | 0.52 ± 0.02 | 0.53 ± 0.03 | 0.55 ± 0.02 |
| BWCFR (WM; $\alpha = 1.0$) | 0.97 ± 0.04 | 0.96 ± 0.06 | 0.95 ± 0.03 | 0.95 ± 0.02 | 0.58 ± 0.10 | 0.54 ± 0.03 | 0.53 ± 0.02 | 0.51 ± 0.02 |
| Oracle | | | | 0.457 | | | | |

Lower = better (best in bold, second best underlined)
Results for $k$-NN, BART, and C-Forest are duplicated for different $d_\phi$

| | | $\text{rPEHE}_{\text{out}}$ | | | | | | |
|---|---|---|---|---|---|---|---|---|
| $d_\phi$ | | 1 | | | | 2 | | |
| $n_{\text{train}}$ | 500 | 1000 | 5000 | 10000 | 500 | 1000 | 5000 | 10000 |
| $k$-NN | **0.57 ± 0.03** | **0.55 ± 0.03** | **0.51 ± 0.01** | 0.51 ± 0.02 | 0.57 ± 0.03 | 0.55 ± 0.03 | **0.51 ± 0.01** | 0.51 ± 0.02 |
| BART | 0.64 ± 0.03 | 0.62 ± 0.04 | 0.52 ± 0.02 | 0.50 ± 0.02 | 0.64 ± 0.03 | 0.62 ± 0.04 | 0.52 ± 0.02 | 0.50 ± 0.02 |
| C-Forest | 0.68 ± 0.03 | 0.61 ± 0.03 | 0.51 ± 0.02 | 0.49 ± 0.01 | 0.68 ± 0.03 | 0.61 ± 0.03 | 0.51 ± 0.02 | 0.49 ± 0.01 |
| TARNet | 0.95 ± 0.05 | 0.91 ± 0.07 | 0.89 ± 0.08 | 0.92 ± 0.05 | 0.59 ± 0.11 | 0.59 ± 0.07 | 0.55 ± 0.05 | 0.57 ± 0.04 |
| BNN (MMD; $\alpha = 0.1$) | 0.97 ± 0.03 | 0.98 ± 0.06 | 0.98 ± 0.02 | 0.98 ± 0.03 | 0.70 ± 0.10 | 0.65 ± 0.11 | 0.64 ± 0.08 | 0.60 ± 0.09 |
| CFR (MMD; $\alpha = 0.1$) | 0.95 ± 0.03 | 0.97 ± 0.05 | 0.93 ± 0.06 | 0.96 ± 0.02 | 0.62 ± 0.07 | 0.62 ± 0.08 | 0.62 ± 0.06 | 0.59 ± 0.08 |
| CFR (MMD; $\alpha = 0.5$) | 0.96 ± 0.03 | 0.99 ± 0.04 | 0.97 ± 0.02 | 0.98 ± 0.02 | 0.66 ± 0.08 | 0.72 ± 0.07 | 0.67 ± 0.09 | 0.67 ± 0.08 |
| CFR (WM; $\alpha = 1.0$) | 0.95 ± 0.03 | 0.97 ± 0.05 | 0.95 ± 0.02 | 0.95 ± 0.02 | 0.56 ± 0.05 | 0.57 ± 0.06 | 0.61 ± 0.08 | 0.54 ± 0.04 |
| CFR (WM; $\alpha = 2.0$) | 0.95 ± 0.03 | 0.97 ± 0.05 | 0.97 ± 0.05 | 0.97 ± 0.02 | 0.58 ± 0.07 | 0.61 ± 0.06 | 0.58 ± 0.09 | 0.57 ± 0.07 |
| InvTARNet | 0.89 ± 0.04 | 0.90 ± 0.06 | 0.87 ± 0.05 | 0.89 ± 0.06 | **0.53 ± 0.04** | **0.52 ± 0.03** | **0.51 ± 0.03** | 0.50 ± 0.02 |
| RCFR (WM; $\alpha = 1.0$) | 0.96 ± 0.04 | 0.98 ± 0.07 | 0.96 ± 0.03 | 0.96 ± 0.03 | 0.57 ± 0.04 | 0.55 ± 0.04 | 0.56 ± 0.03 | 0.56 ± 0.04 |
| CFR-ISW (WM; $\alpha = 1.0$) | 0.96 ± 0.03 | 0.97 ± 0.05 | 1.00 ± 0.10 | 0.97 ± 0.02 | 0.54 ± 0.04 | 0.53 ± 0.02 | 0.53 ± 0.03 | 0.55 ± 0.02 |
| BWCFR (WM; $\alpha = 1.0$) | 0.96 ± 0.04 | 0.96 ± 0.05 | 0.96 ± 0.04 | 0.95 ± 0.03 | 0.58 ± 0.10 | 0.54 ± 0.04 | 0.53 ± 0.03 | 0.52 ± 0.03 |
| Oracle | | | | 0.457 | | | | |

Lower = better (best in bold, second best underlined)
Results for $k$-NN, BART, and C-Forest are duplicated for different $d_\phi$

Table 7: Results for IHDP100 experiments. Reported: in-sample and out-sample rPEHE; mean over 100 train/test splits.

| | rPEHE$_{\text{in}}$ | | | | |
|---|---|---|---|---|---|
| $d_\phi$ | 5 | 10 | 15 | 20 | 25 |
| $k$-NN | $0.668 \pm 0.064$ | $0.668 \pm 0.064$ | $0.668 \pm 0.064$ | $0.668 \pm 0.064$ | $0.668 \pm 0.064$ |
| BART | $0.594 \pm 0.124$ | $0.594 \pm 0.124$ | $0.594 \pm 0.124$ | $0.594 \pm 0.124$ | $0.594 \pm 0.124$ |
| C-Forest | $0.641 \pm 0.066$ | $0.641 \pm 0.066$ | $0.641 \pm 0.066$ | $0.641 \pm 0.066$ | $0.641 \pm 0.066$ |
| TARNet | $0.514 \pm 0.245$ | $0.503 \pm 0.235$ | $0.502 \pm 0.236$ | $0.505 \pm 0.232$ | $0.503 \pm 0.234$ |
| BNN (MMD; $\alpha = 0.1$) | $0.495 \pm 0.172$ | $0.490 \pm 0.175$ | $0.500 \pm 0.173$ | $0.498 \pm 0.172$ | $\underline{0.469 \pm 0.184}$ |
| CFR (MMD; $\alpha = 0.1$) | $\underline{0.463 \pm 0.186}$ | $0.470 \pm 0.191$ | $0.477 \pm 0.181$ | $\underline{0.469 \pm 0.188}$ | $0.476 \pm 0.183$ |
| CFR (MMD; $\alpha = 0.5$) | $\mathbf{0.462 \pm 0.185}$ | $0.489 \pm 0.190$ | $0.503 \pm 0.185$ | $0.492 \pm 0.176$ | $0.500 \pm 0.185$ |
| CFR (WM; $\alpha = 1.0$) | $\underline{0.463 \pm 0.190}$ | $\mathbf{0.465 \pm 0.195}$ | $0.467 \pm 0.190$ | $\mathbf{0.466 \pm 0.192}$ | $\mathbf{0.466 \pm 0.194}$ |
| CFR (WM; $\alpha = 2.0$) | $0.471 \pm 0.183$ | $0.473 \pm 0.191$ | $0.475 \pm 0.189$ | $0.471 \pm 0.191$ | $0.471 \pm 0.188$ |
| InvTARNet | $0.505 \pm 0.258$ | $0.537 \pm 0.310$ | $0.523 \pm 0.240$ | $0.514 \pm 0.235$ | $0.505 \pm 0.226$ |
| RCFR (WM; $\alpha = 1.0$) | $0.560 \pm 0.154$ | $0.572 \pm 0.147$ | $0.499 \pm 0.233$ | $0.653 \pm 0.253$ | $0.506 \pm 0.205$ |
| CFR-ISW (WM; $\alpha = 1.0$) | $0.466 \pm 0.183$ | $\underline{0.469 \pm 0.189}$ | $\mathbf{0.464 \pm 0.195}$ | $\mathbf{0.466 \pm 0.195}$ | $0.471 \pm 0.199$ |
| BWCFR (WM; $\alpha = 1.0$) | $0.581 \pm 0.163$ | $\underline{0.600 \pm 0.166}$ | $0.599 \pm 0.175$ | $0.595 \pm 0.204$ | $0.789 \pm 1.721$ |
| Oracle | 0.425 | | | | |

Lower = better (best in bold, second best underlined)
Results for $k$-NN, BART, and C-Forest are duplicated for different $d_\phi$

| | rPEHE$_{\text{out}}$ | | | | |
|---|---|---|---|---|---|
| $d_\phi$ | 5 | 10 | 15 | 20 | 25 |
| $k$-NN | $0.756 \pm 0.123$ | $0.756 \pm 0.123$ | $0.756 \pm 0.123$ | $0.756 \pm 0.123$ | $0.756 \pm 0.123$ |
| BART | $0.615 \pm 0.133$ | $0.615 \pm 0.133$ | $0.615 \pm 0.133$ | $0.615 \pm 0.133$ | $0.615 \pm 0.133$ |
| C-Forest | $0.667 \pm 0.123$ | $0.667 \pm 0.123$ | $0.667 \pm 0.123$ | $0.667 \pm 0.123$ | $0.667 \pm 0.123$ |
| TARNet | $0.620 \pm 0.292$ | $0.622 \pm 0.288$ | $0.620 \pm 0.291$ | $0.613 \pm 0.285$ | $0.613 \pm 0.285$ |
| BNN (MMD; $\alpha = 0.1$) | $0.514 \pm 0.181$ | $0.510 \pm 0.185$ | $0.517 \pm 0.183$ | $0.515 \pm 0.181$ | $\underline{0.496 \pm 0.195}$ |
| CFR (MMD; $\alpha = 0.1$) | $\mathbf{0.492 \pm 0.193}$ | $\underline{0.497 \pm 0.201}$ | $0.500 \pm 0.186$ | $\mathbf{0.493 \pm 0.191}$ | $0.500 \pm 0.190$ |
| CFR (MMD; $\alpha = 0.5$) | $\underline{0.493 \pm 0.191}$ | $\underline{0.502 \pm 0.191}$ | $0.515 \pm 0.191$ | $0.509 \pm 0.185$ | $0.511 \pm 0.187$ |
| CFR (WM; $\alpha = 1.0$) | $0.499 \pm 0.202$ | $0.500 \pm 0.211$ | $\mathbf{0.495 \pm 0.203}$ | $\underline{0.497 \pm 0.204}$ | $0.498 \pm 0.204$ |
| CFR (WM; $\alpha = 2.0$) | $0.500 \pm 0.198$ | $\mathbf{0.496 \pm 0.201}$ | $0.503 \pm 0.202$ | $\underline{0.499 \pm 0.203}$ | $0.500 \pm 0.196$ |
| InvTARNet | $0.563 \pm 0.269$ | $0.610 \pm 0.350$ | $0.618 \pm 0.287$ | $0.605 \pm 0.277$ | $0.595 \pm 0.267$ |
| RCFR (WM; $\alpha = 1.0$) | $0.601 \pm 0.202$ | $0.612 \pm 0.198$ | $0.535 \pm 0.228$ | $0.676 \pm 0.281$ | $0.580 \pm 0.256$ |
| CFR-ISW (WM; $\alpha = 1.0$) | $0.501 \pm 0.195$ | $0.501 \pm 0.203$ | $\underline{0.496 \pm 0.207}$ | $\underline{0.497 \pm 0.208}$ | $\mathbf{0.495 \pm 0.210}$ |
| BWCFR (WM; $\alpha = 1.0$) | $0.599 \pm 0.182$ | $0.618 \pm 0.190$ | $0.604 \pm 0.184$ | $0.601 \pm 0.190$ | $0.731 \pm 0.988$ |
| Oracle | 0.424 | | | | |

Lower = better (best in bold, second best underlined)
Results for $k$-NN, BART, and C-Forest are duplicated for different $d_\phi$

Table 8: Results for HC-MNIST experiments. Reported: in-sample and out-sample rPEHE; mean over 10 runs.

| | rPEHE$_{\text{in}}$ | | | rPEHE$_{\text{out}}$ | | |
|---|---|---|---|---|---|---|
| $d_\phi$ | 7 | 39 | 78 | 7 | 39 | 78 |
| $k$-NN | $1.06 \pm 0.01$ | $1.06 \pm 0.01$ | $1.06 \pm 0.01$ | $1.10 \pm 0.01$ | $1.10 \pm 0.01$ | $1.10 \pm 0.01$ |
| BART | $0.89 \pm 0.24$ | $0.89 \pm 0.24$ | $0.89 \pm 0.24$ | $0.89 \pm 0.25$ | $0.89 \pm 0.25$ | $0.89 \pm 0.25$ |
| C-Forest | $0.82 \pm 0.00$ | $0.82 \pm 0.00$ | $0.82 \pm 0.00$ | $0.82 \pm 0.00$ | $0.82 \pm 0.00$ | $0.82 \pm 0.00$ |
| TARNet | $\mathbf{0.71 \pm 0.02}$ | $\mathbf{0.66 \pm 0.01}$ | $\mathbf{0.67 \pm 0.03}$ | $\mathbf{0.72 \pm 0.02}$ | $\mathbf{0.70 \pm 0.01}$ | $\mathbf{0.71 \pm 0.02}$ |
| BNN (MMD; $\alpha = 0.1$) | $0.75 \pm 0.03$ | $0.73 \pm 0.03$ | $0.93 \pm 0.14$ | $0.77 \pm 0.03$ | $0.74 \pm 0.03$ | $0.93 \pm 0.14$ |
| CFR (MMD; $\alpha = 0.1$) | $\underline{0.73 \pm 0.03}$ | $\underline{0.67 \pm 0.01}$ | $\underline{0.69 \pm 0.03}$ | $\underline{0.74 \pm 0.02}$ | $\underline{0.71 \pm 0.01}$ | $\underline{0.73 \pm 0.02}$ |
| CFR (MMD; $\alpha = 0.5$) | $0.86 \pm 0.23$ | $0.74 \pm 0.02$ | $0.80 \pm 0.15$ | $0.86 \pm 0.13$ | $0.76 \pm 0.02$ | $0.78 \pm 0.05$ |
| CFR (WM; $\alpha = 1.0$) | $1.10 \pm 0.17$ | $1.21 \pm 0.14$ | $1.37 \pm 0.23$ | $1.10 \pm 0.17$ | $1.19 \pm 0.14$ | $1.28 \pm 0.14$ |
| CFR (WM; $\alpha = 2.0$) | $1.40 \pm 0.31$ | $1.33 \pm 0.09$ | $1.30 \pm 0.05$ | $1.28 \pm 0.10$ | $1.30 \pm 0.06$ | $1.29 \pm 0.06$ |
| InvTARNet | $0.77 \pm 0.05$ | $0.69 \pm 0.04$ | $\underline{0.69 \pm 0.05}$ | $0.76 \pm 0.03$ | $\underline{0.71 \pm 0.01}$ | $\mathbf{0.71 \pm 0.01}$ |
| RCFR (WM; $\alpha = 1.0$) | $1.83 \pm 1.48$ | $2.78 \pm 4.05$ | $3.68 \pm 3.43$ | $1.24 \pm 0.29$ | $2.66 \pm 4.33$ | $3.05 \pm 3.72$ |
| CFR-ISW (WM; $\alpha = 1.0$) | $1.29 \pm 0.12$ | $1.25 \pm 0.04$ | $1.28 \pm 0.06$ | $1.25 \pm 0.05$ | $1.26 \pm 0.08$ | $1.27 \pm 0.07$ |
| BWCFR (WM; $\alpha = 1.0$) | $2.92 \pm 2.39$ | $1.31 \pm 0.05$ | $1.35 \pm 0.05$ | $1.46 \pm 0.45$ | $1.32 \pm 0.06$ | $1.35 \pm 0.05$ |
| Oracle | 0.522 | | | 0.513 | | |

Lower = better (best in bold, second best underlined)
Results for $k$-NN, BART, and C-Forest are duplicated for different $d_\phi$

### E.2 POLICY ERROR RATES AND DEFERRAL RATES

Here, we examine the trade-off between the improvement in the error rates and the increase in the deferral rates, which appear after the application of our refutation framework. One way to do so is through *policy error rate vs. deferral rate* plots, as in (Jesson et al., 2021). In Figures 4, 5, and 6, we report the error rates and the deferral rates for synthetic data, IHDP100 dataset, and HC-MNIST dataset, respectively. Therein, the baseline representation learning methods for CATE are shown as points located at the vertical line with a deferral rate of zero. Each of those points is then connected with a point corresponding to our bounds performance. As a result, we see that our refutation framework achieves an improvement in the error rates with only a marginal increase in the deferral rates.

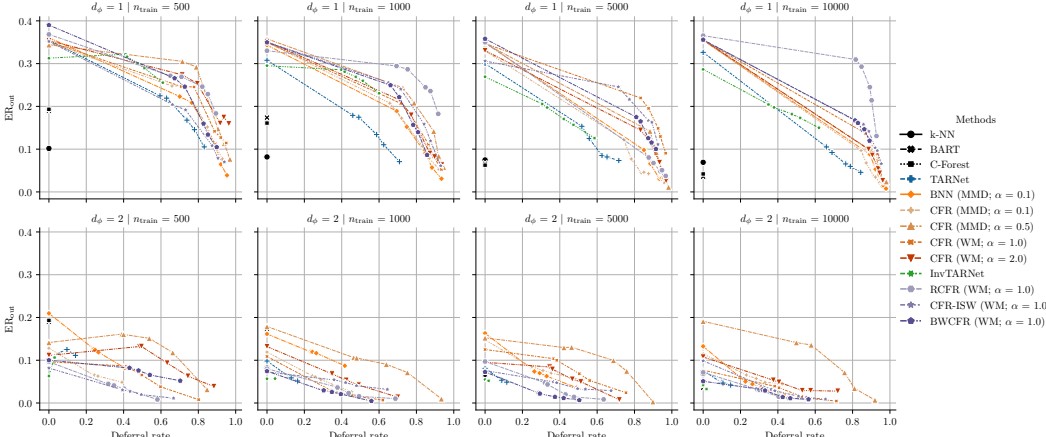

Figure 4: Policy error rate vs. deferral rate plot for synthetic data. Reported: out-sample performance of baseline methods connected with the performance of our refutation framework; mean over 10 runs. Here, $\delta$ varies in the range $\{0.0005, 0.001, 0.005, 0.01, 0.05\}$, which corresponds to several scatter points per line.

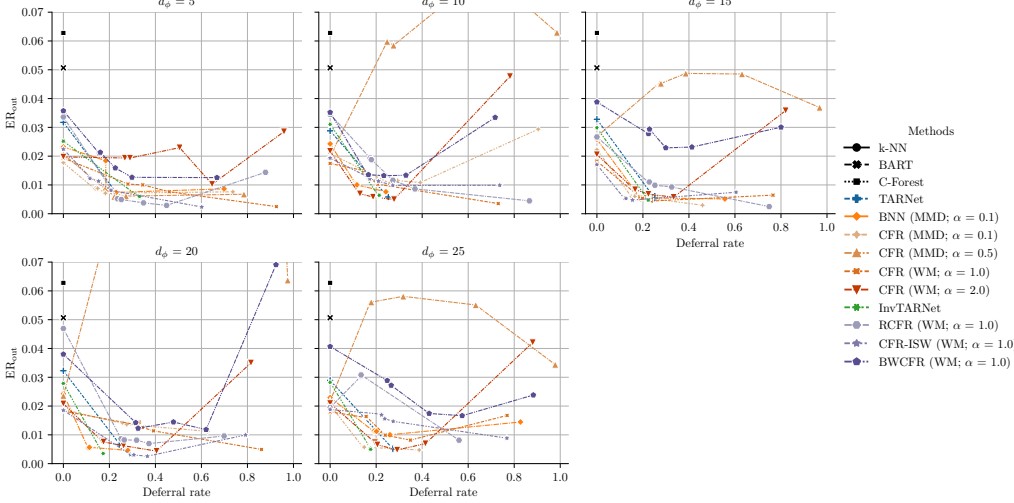

Figure 5: Policy error rate vs. deferral rate plot for IHDP100 dataset. Reported: out-sample performance of baseline methods connected with the performance of our refutation framework; mean over 100 train/test splits. Here, $\delta$ varies in the range $\{0.0005, 0.001, 0.005, 0.01, 0.05\}$, which corresponds to several scatter points per line.

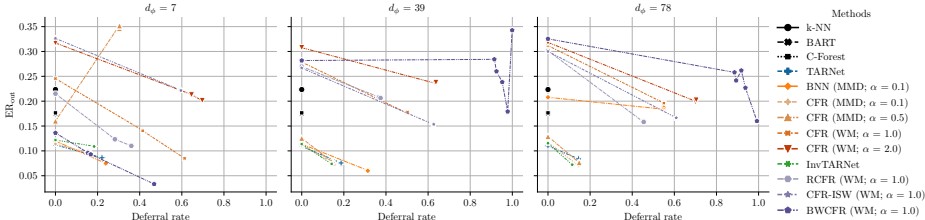

Figure 6: Policy error rate vs. deferral rate plot for HC-MNIST dataset. Reported: out-sample performance of baseline methods connected with the performance of our refutation framework; mean over 10 runs. Here, $\delta$ varies in the range $\{0.0005, 0.001, 0.005, 0.01, 0.05\}$, which corresponds to several scatter points per line.

## E.3 DECISION BOUNDARIES

In Fig. 7 and 8, we provide plots of the decision boundaries for all the baselines (using the synthetic benchmark). Therein, we plot both the ground-truth boundaries and boundaries, estimated with our refutation framework in combination with different representation learning methods for CATE. On the left side ($d_\phi = 1$), the low-dimensional representation induces at the same time loss of heterogeneity and confounding bias. We see that our bounds remain approximately valid for CATE wrt. representations, even when the dimensionality of the representation is misspecified, especially for different variants of CFR and InvTARNet. On the right side ($d_\phi = 2$), there is no loss of heterogeneity but still some hidden confounding induced. Yet, our bounds are valid for both CATE wrt. representations and covariates for all the baselines. Additionally, we observe that the tightness of the estimated bounds highly depends on the representation learning baseline, e. g., the bounds on the RICB are the tightest for InvTARNet, as its representations introduce almost no confounding bias.

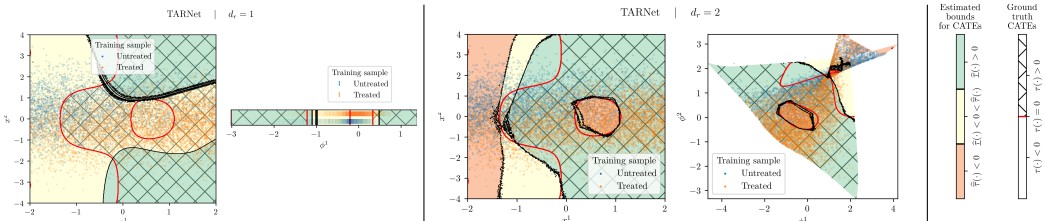

Figure 7: Decision boundaries for the synthetic dataset. Ground-truth and estimated boundaries are shown with hatches and colors, respectively. Training samples ($n_{\text{train}} = 10,000$) are shown too to highlight treatment imbalance. Here, $\delta = 0.01$.

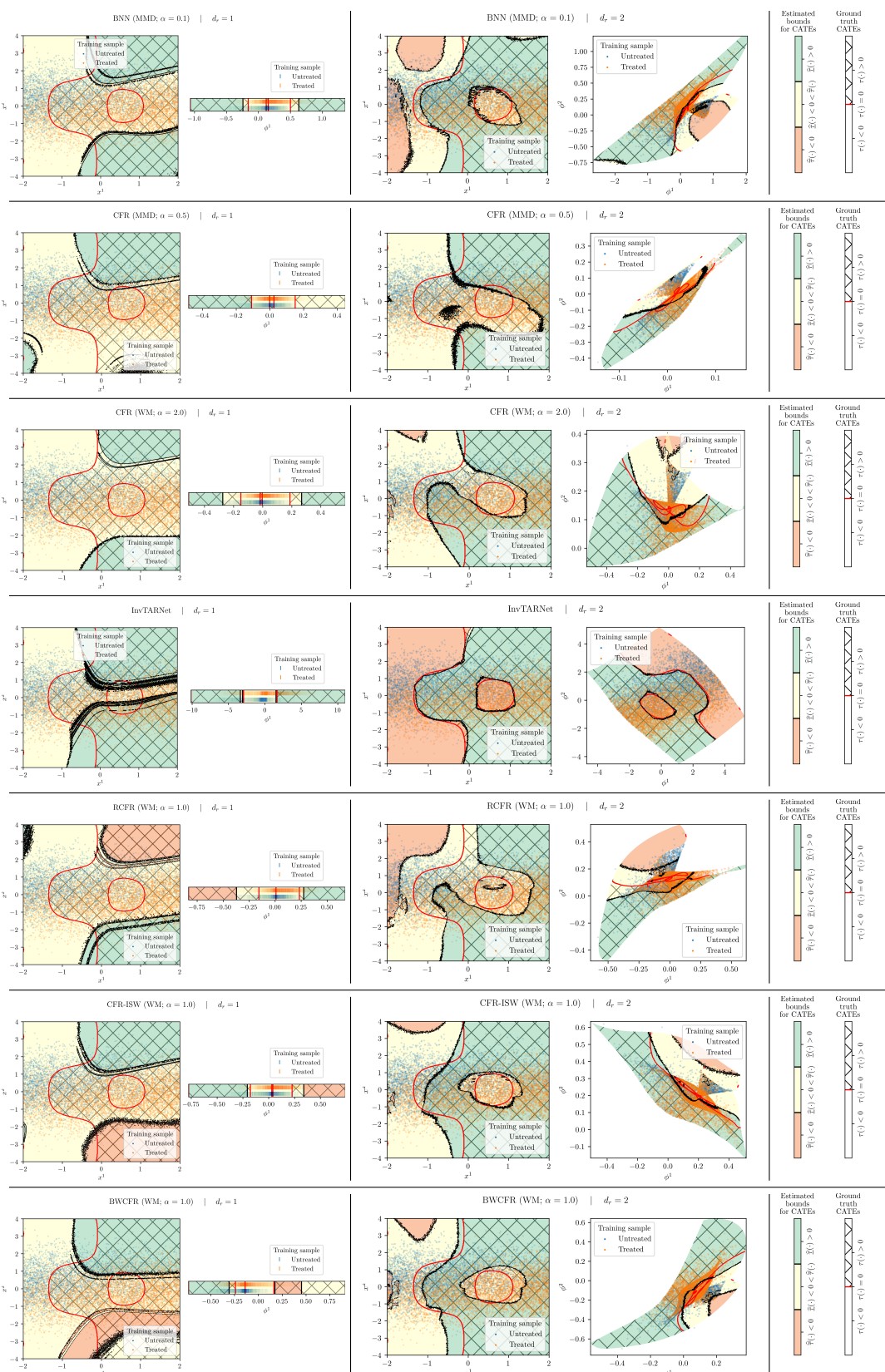

Figure 8: Decision boundaries for the synthetic dataset. Ground-truth and estimated boundaries are shown with hatches and colors, respectively. Training samples ($n_{\text{train}} = 10,000$) are shown too to highlight treatment imbalance. Here, $\delta = 0.01$.

