# OpenReview forum: "Bounds on Representation-Induced Confounding Bias for Treatment Effect Estimation"
_ICLR.cc/2024/Conference — ICLR 2024 spotlight_

### Official Review · Reviewer_FevT · 2023-10-27

**Soundness:** 2 fair
**Presentation:** 2 fair
**Contribution:** 2 fair
**Rating:** 8
**Confidence:** 3

**Summary:**

This paper discusses a setting where a representation function $\phi(X)$, (a generalization of propensity score $\pi(X)$), is available while part of X is unobservable. That is, instead of following the typical approach of choosing X from only observables (expecting $\phi(X)$ to be a balancing score) and discussing the potential effects of unobservable covariates, they follow the approach of considering X as all covariates including even unobservable covariates. At the same time, they assume that $\phi(X)$ value is available (while part of X is not observable)

In the typical approach, when we cannot assume that $\phi(X)$ is a balancing score, we may suffer confounding bias. In the exact same way in their approach, when we cannot assume that $\phi(X)$ (in their case it is called the representation function) includes enough information about unobservable covariates, we may suffer a bias. (in their case it is called the representation-induced confounding bias or RICB)

The authors identify RICB, and propose a technique to estimate the bound.
Comprehensive simulation studies were followed.


---------------------------------------------------------------------------------------------------------------------------------------------
**Replying to the public discussion***

Title: "2021 NeurIPS paper ... assumes that all confounders are X" is what I am saying.

Hi Alicia,

Thank you for participating in the discussion.
Yes, your paper assumes that all confounders are included in X, which is observed. This is your paper's assumption 1, which states that there are no unobserved confounders.

On the other hand, this paper's setting is different.
I quote myself above:
"This paper discusses a setting where a representation function $\phi(X)$, (a generalization of propensity score $\pi(X)$), is available while part of X is unobservable. That is, instead of following the typical approach of choosing X from only observables (expecting $\phi(X)$ to be a balancing score) and discussing the potential effects of unobservable covariates, they follow the approach of considering X as all covariates including even unobservable covariates. At the same time, they assume that $\phi(X)$ value is available (while part of X is not observable)"

My concern is that, the authors are saying that their setting is making a typical assumption (which is not true) and then citing your work as the one of the works that make typical assumption (which is true).

Thanks,
Reviewer FevT.

---------------------------------------------------------------------------------------------------------------------------------------------
**Replying to the public discussion 2***

Hi Alicia,

I understand what you are saying - I think what AC pointed out is correct.
I modified my score, as I have no further concerns.

Thank you very much again for participating in the discussion.

Many thanks,
Reviewer FevT

**Strengths:**

Simulation studies are quite comprehensive.
Theoretical bounds has been proposed.
The paper is very well written. It was pleasant to read.

**Weaknesses:**

\textbf{1. Motivation of their approach}

As discussed in the Summary part of this review, for me it was hard to understand why we need a new approach of choosing X. The concept of RICB is, in essence, equivalent to confounder bias but formulated in a different choice of X. For example, in the traditional way of choosing X as only unobservables and talking about $\phi(X)$ not being a balancing score, potential effect of unobservable covariates not being included as X can be discussed. So I am not sure about the potential benefit of choosing X to include unobservable covariates.

\textbf{2. Bounds}
Theoretical bounds provided should be appreciated, but I cannot be sure how strong this theoretical bound is only from current version of the manuscript.

**Questions:**

In terms of Weakness 1: Could you please give a clear motivation of choosing X to include unobservable but considering $\phi(X)$ as observable, and then reframing the confounder bias we know as RICB in the newly proposed setting? Does the fact that we are dealing with representation learning make some difference? I just want to try to understand.

In terms of Weakness 2: How tight is the bound for some popular special cases, especially for the settings you did your experiment? Are they practically good?

**Details Of Ethics Concerns:**

Deleted.

---

> ### Author Response · Authors · 2023-11-18
> **Response to Reviewer FevT [1 / 2]**
>
> Thank you! We appreciate your valuable review and that you found our paper well-written, and experiments comprehensive. In the following, we will respond to your questions.
>
> **Response to Weaknesses and Questions.**
>
>
>
> 1. **Motivation of our approach.** We are more than happy to clarify the motivation behind our approach.
>
>   + _What is the motivation behind our approach?_ In our paper, we build upon the task of CATE estimation. That is, we follow previous literature [1-7] and assume that **all the confounders** are **observed** inside of $X$. Hence, when using representation learning methods, one maps an $X$ onto lower-dimensional representation $\Phi(X)$. Formally, $X$ is $n$-dimensional and $\Phi(X)$ is $p$-dimensional with $p &lt; n$. Then, by learning the representation $\Phi(X)$, we face an information loss as not all information about $X$ can be stored in the representation. In simple words, $X$ is now split into (i) the main component $\Phi(X)$ which is used for the downstream analysis and (ii) an implicit (complementary) component $\Omega(X)$ that captures everything else. Importantly, not only $X$ is observed, but also both components $\Phi(X)$ and $\Omega(X)$ are observed or could be inferred. However, CATE estimation using representation learning is only based on component (i) and not component (ii), (which is discarded). In the language of our paper, **component (ii) is all the information, discarded by the representation, which could contain ground-truth confounders and, thus, lead to bias**.
>
>
>    +  _How is our approach related to traditional, hidden confounders?_ An alternative view of component (ii) is to build upon causal sensitivity analysis and simply say that these are essentially ‘non-observable’ confounders. The reason is that, even though they are observed, they are actually not used in existing CATE estimation methods. As a result, **CATE estimation with a lower-dimensional representation leads to a confounding bias**. We highlight the differences between (a) traditional, hidden confounding and (b) our setting in the following table.
>   | Setting / Application            | Observed variables | Unobserved variables | Ignorability            | Confounding induced by the usage of |
>   |----------------------------------|--------------------|----------------------|-------------------------|-------------------------------------|
>   | Hidden confounding               | $X$                | $U$                  | $Y[a] \mathrel{\unicode{x2AEB}} A \mid X, U$ | $X$                                 |
>   | Representation learning for CATE | $X, \Phi(X)$       | —                    | $Y[a] \mathrel{\unicode{x2AEB}} A \mid X$    | $\Phi(X)$                           |
>
>    + _Why is our approach relevant and general?_ As we show in our paper, such bias is inherent to **almost all** existing CATE methods that build upon representation learning. This just happens naturally because learning a lower-dimensional representation leads to information loss and thus confounding bias. Oftentimes, such confounding bias is reinforced by the learning objective (e.g., balancing), which presents further sources of bias. Of note, there is no ‘easy fix’ (or ‘no free lunch’) to the confounding bias from representation learning. On the one hand, CATE based on representation learning is actively searching to learn a lower-dimensional representation as this has lower variance and generalizes better. So, we actively aim for $p &lt; n$ (and not $p \approx n$). Yet, this leads to confounding bias. On the other hand, we could remove confounding bias by setting $p \approx n$, yet this would preclude any gains in prediction performance from representation learning.
>
>        To avoid confusion, we would also like to clarify the following. The bidirectional edge in Fig. 1 between $Y$ and $X$ is not used to indicate hidden confounding wrt. to treatment $A$, but rather to denote the general causal diagram, implied by assumptions (1)-(3).
>
> &nbsp; &nbsp; &nbsp;  **Action:** We added clarifications along the above lines to our paper.

---

> ### Author Response · Authors · 2023-11-18
> **Response to Reviewer FevT [2 / 2]**
>
> 2. **Bounds.** Thank you for raising this important question. We added a **new Corrolary 1** in **Appendix B,** where we state that bounds are **valid and sharp** (in the sense of sensitivity models). This result was formally proved in [8]. This implies the following:
>    * Valid means that our bounds always contain the ground-truth representation CATE (given the ground-truth representation-conditional distribution of the outcome).
>    * Sharp means that the bounds only include the observational distributions complying with a sensitivity constraint from Eq. 10.
>
>     We empirically evaluated validity and sharpness jointly. E.g., the reported results in Tables 2-4 and Figures 4-6 showed improvements over the baselines due to our framework. Therein, a drop in error rates implies validity, and the fact that we defer not too many observations/individuals leads to sharpness.
>
>
>     A further question is how tight our bounds are, yet tightness is an empirical quantity that depends on both the fitted CATE estimator and a chosen sensitivity model. Hence, we cannot assess tightness theoretically but only through empirical analysis. To do so, we added new **Figures 7 and 8,** where we plot the ground-truth decision boundaries and decision boundaries based on the estimated bounds for the synthetic benchmark. We see that estimated bounds are valid (they contain the ground-truth decision boundary of CATE wrt. representations) and sufficiently tight, especially, when the representation is of the same dimensionality as the covariates (so there is less chance of the presence of the RICB). This confirms the effectiveness of our bounds.
>
>     **Action:** We added a new **Corrolary 1 **in** Appendix B** on the properties of the MSM bounds to the revised version of the paper. We added **Figures 7-8** where we visualize the bounds (ground-truth and estimated decision boundaries) to show that our bounds are valid and tight.
>
>
>  \
> **References:**
>
> [1] Ahmed M. Alaa and Mihaela van der Schaar. Bayesian inference of individualized treatment effects using multi-task Gaussian processes. In Advances in Neural Information Processing Systems, 2017.
>
> [2] Ahmed M. Alaa and Mihaela van der Schaar. Bayesian nonparametric causal inference: Information rates and learning algorithms. IEEE Journal of Selected Topics in Signal Processing, 12:1031–1046, 2018a.
>
> [3] Ahmed M. Alaa and Mihaela van der Schaar. Limits of estimating heterogeneous treatment effects: Guidelines for practical algorithm design. In International Conference on Machine Learning, 2018b.
>
> [4] Alicia Curth and Mihaela van der Schaar. On inductive biases for heterogeneous treatment effect estimation. Advances in Neural Information Processing Systems, 2021a.
>
> [5] Alicia Curth and Mihaela van der Schaar. Nonparametric estimation of heterogeneous treatment effects: From theory to learning algorithms. In International Conference on Artificial Intelligence and Statistics, 2021b.
>
> [6] Uri Shalit, Fredrik D. Johansson, and David Sontag. Estimating individual treatment effect: Generaliization bounds and algorithms. In International Conference on Machine Learning, 2017.
>
> [7] Fredrik D. Johansson, Uri Shalit, Nathan Kallus, and David Sontag. Generalization bounds and representation learning for estimation of potential outcomes and causal effects. Journal of Machine Learning Research, 23:7489–7538, 2022.
>
> [8] Dennis Frauen, Valentyn Melnychuk, and Stefan Feuerriegel. Sharp bounds for generalized causal sensitivity analysis. In Advances in Neural Information Processing Systems, 2023.

---

> ### Public Comment · ~Alicia_Curth1 · 2023-11-27
> **Correcting a misconception in this review**
>
> Dear Reviewer FevT and Area Chairs,
>
> As the author of the paper referenced in the review above, I would like to correct a clear misconception in this review. (To clarify, I have no involvement with this paper, but became aware of the misconceptions in the review with respect to my work).
>
> I can confidently state that all my work cited in the response by the authors, including and especially the 2021 NeurIPS paper cited in the review above, **assumes that all confounders are included in X** exactly as stated by the authors. This should be obvious from the passage quoted in the review above, which clearly states the assumption "Unconfoundedness: there are no unobserved confounders". If there were any confounders that were not recorded in X, then $Y(0), Y(1) \indep W | X$ would not hold as assumed.
>
> The authors of this paper have thus in **no way** ''tried to support the motivation of their setting by providing a misinformation'' as implied by the Reviewer. I strongly encourage this Reviewer to retract this incorrect statement, and would otherwise suggest to the ACs to disregard this review.
>
> With kind regards,
>
> Alicia Curth

---

> > ### Public Comment · ~Alicia_Curth1 · 2023-11-27
> > **Follow-up**
> >
> > Dear Reviewer FevT,
> >
> > Thank you for getting back on this so quickly -- the clarification of what was meant by the statement in the 'details for ethics review' section in a previous iteration of the top-level review resolves at least some part of the prior misunderstanding!
> >
> > If I may, I would like to emphasise that I think that there is still a misconception in the original review with respect to what the assumption is in the paper under review here. As far as I can tell, this paper *does not assume that there are hidden confounders included in X*. To the contrary, the paper assumes that if we were to condition on everything we have observed -- i.e.  X itself -- unconfoundedness holds (see page 4). This is indeed the standard assumption made in most of the ML CATE estimation literature. (The statement "At the same time, they assume that $\Phi(X)$ value is available (while part of X is not observable)" from the original review thus does not seem to be correct to me.)
> >
> > Instead, the reason that there might be RICB is not that part of $X$ is not available, but that part of $X$ *is not used* by some ML estimators. This is because a popular approach in the ML literature on CATE estimation since Shalit et al (2017) has been to then _learn_ a representation function $\Phi(X)$ of this possibly high-dimensional X to be used by the downstream CATE estimator. In the paper under review, the authors demonstrate that a consequence of _learning_ non-invertible representations of an X (that is itself fully observed) using such  popular ML methods can -- perhaps counterintuitively -- be that *previously not present confounding bias* is now induced by the loss of information through the learned representation. (This would be the case e.g. if  balancing representations as proposed in Shalit et al (2017) were learned by CFRNet with regularisation penalty set too high). That RICB can occur is thus not a 'non-standard assumption', it is simply a fact: if the regularisation penalty for the balancing representations is set to infinity, CFRNet will learn a representation $\Phi(X)\equiv\Phi$ that is constant and equal across the two treatment groups to minimise the distance in representation space, which will induce RICB.
> >
> > I hope this helps in clearing up some confusions.
> >
> > Best wishes,
> > Alicia

---

### Official Review · Reviewer_8op8 · 2023-10-31

**Soundness:** 3 good
**Presentation:** 2 fair
**Contribution:** 3 good
**Rating:** 8
**Confidence:** 2

**Summary:**

This paper tackles the problem of confounding bias induced by learning representation of confounder for CATE estimation. The authors proposed a framework for estimating bounds on the induced confounding bias. A neural framework is used to compute the bounds.

**Strengths:**

- The paper presents a problem that is novel and related to the representation of learning for CATE, which is a prominent research direction.
- A detailed analysis of representation-induced bias is provided.
- Both real-world and synthetic experiments are performed with the proposed framework.

**Weaknesses:**

- The motivation for employing CDAG is not quite clear.
- No theoretical proof of the proposed bounds.

**Questions:**

- Could you provide some intuition about learning the representation of all the covariates together instead of the confounder?
- If learning representation of covariates inducing bias is unavoidable, how does the bias compare with bias due to finite-sample? e.g., How does it compare with the non-representation learning approach?

---

> ### Author Response · Authors · 2023-11-18
> **Response to Reviewer 8op8**
>
> Thank you for your review. We are pleased that you found our framework novel and the research direction prominent. We would like to elaborate on the mentioned weaknesses and questions.
>
> **Answer to Weaknesses:**
>
> * Thank you for asking this question. We employ a clustered DAG [1] for **a better intuition about valid and invalid representations**. With the clustered DAG, we do not need to know the exact structure of all the sub-covariates and can instead provide a more abstract, clustered causal diagram. This is a helpful way to describe the valid adjustment set [2], i.e., all the covariates are split into four categories of (1) noise, (2) instruments, (3) outcome-predictive covariates, and (4) confounders. Importantly, the exact partitioning of covariates is unknown in practice, and our MSM framework does not assume any specific partitioning.
>
>     **Action:** We clarified in our paper that we adopt a clustered DAG only for intuition purposes.
>
> * Thank you for suggesting to make the derivation of the bounds more explicit. For our bounds, we adapted the theoretic results, provided in [3,4]. We followed your suggestion and added a more rigorous derivation (see new **Lemma 3** and the theoretical proof).
>
>     **Action:** We added a new **Lemma 3** to **Appendix B** where we detail the derivation of the bounds.
>
>
> **Answer to Questions:**
>
>
>
> * Thank you. In real-world applications, the ground-truth partitioning or ground-truth confounders are usually unknown. Therefore, representation learning methods for CATE work with datasets that typically contain as many potentially useful covariates as possible.
>
>     **Action:** We added to the revised version of the paper that representation learning methods for CATE do not assume a specific partitioning and consider all the covariates as input. This typically helps in the downstream performance.
>
> * The question about the trade-off between finite-sample bias and representation-induced confounding bias is an interesting direction for future theoretic research. We could envision a work where such tradeoffs are learned explicitly but there are several open challenges (e.g., how do you accurately estimate the finite-sample bias? While Bayesian methods are a natural approach, this typically precludes the use of neural methods, etc.).
>
>    Nevertheless, our paper is, to the best of our knowledge, the **first** work to raise the question of induced confounding in the representations, which would be important for any future research. Hence, we hope that our work spurs further follow-up research.
>
>    Regarding the relevance to non-representation learning CATE estimators: we followed your suggestion and added non-neural CATE  baselines (k-NN, BART, and Causal Forests) to our revised manuscript. These are in our revised **Tables 2-8 and Figures 4-6**. In the same way as for other baselines, we evaluated rPEHE and the error rate of policies based on estimated CATE. In the new results, we observed that, e.g., non-neural methods outperformed the representation learning CATE estimators on the simple synthetic benchmark with large sample sizes, but fell short in other more complex scenarios (e.g., benchmarks with high-dimensional covariates or low-sample sizes).
>
>     **Action:** We added three non-neural CATE baselines, i.e., k-NN, BART, and Causal Forests to the revised version of the paper. See our revised **Tables 2-8** and **Figures 4-6**.
>
>
> **References:**
>
> [1] Tara V. Anand, Adele H. Ribeiro, Jin Tian, and Elias Bareinboim. Causal effect identification in cluster DAGs. In AAAI Conference on Artificial Intelligence, 2023.
>
> [2] Carlos Cinelli, Andrew Forney, and Judea Pearl. A crash course in good and bad controls. Sociological Methods & Research, 2022.
>
> [3] Miruna Oprescu, Jacob Dorn, Marah Ghoummaid, Andrew Jesson, Nathan Kallus, and Uri Shalit. B-learner: Quasi-oracle bounds on heterogeneous causal effects under hidden confounding. In International Conference on Machine Learning, 2023.
>
> [4] Dennis Frauen, Valentyn Melnychuk, and Stefan Feuerriegel. Sharp bounds for generalized causal sensitivity analysis. In Advances in Neural Information Processing Systems, 2023.

---

> > ### Comment · Reviewer_8op8 · 2023-12-04
> >
> > Thanks for the clarification. After reading the revision and other reviews, I think all my concerns have been addressed. I have updated my evaluation score.

---

### Official Review · Reviewer_Pw4L · 2023-10-31

**Soundness:** 2 fair
**Presentation:** 2 fair
**Contribution:** 2 fair
**Rating:** 5
**Confidence:** 4

**Summary:**

Estimating conditional average treatment effect (CATE) estimation widely uses low-dimensional representation learning, which can lose information about the observed confounders and thus lead to bias.
In this paper, the authors propose a new framework for estimating bounds on the representation-induced confounding bias (RICB). To summarize, the contributions are three-fold:
1.	CATE from representation learning methods can be non-identifiable due to RICB.
2.	The authors propose a representation-agnostic framework to perform partial identification of CATE.
3.	The authors demonstrate the effectiveness of our bounds together with a wide range of state-of-the-art CATE methods.

**Strengths:**

The paper is technically sound and well-organized.

**Weaknesses:**

It seems that the notations/symbols are not defined correctly. For example, in the section of notations, the authors claim that $\mu_a^x(x)=\mathbb{E}(Y|A=1,X=x)$, but $\mu_a^x(x)$ should be $\mathbb{E}(Y|A=a,X=x)$. In the same paragraph, the authors claim that $\mu_a^\phi(\phi)=\mathbb{E}(Y|A=1,\Phi(X)=\phi)$, but $\mu_a^\phi(\phi)$ should be $\mathbb{E}(Y|A=a,\Phi(X)=\phi)$. In addition, the authors define $\pi_a^x(x)= \mathbb{P}(A=a|X=x)$. I wonder why the authors do not simply $\pi_a^x$ or $\pi_a(x)$. Problem arises when the authors introduce overlap assumption. The authors claim that $\mathbb{P}(0<\pi_a^x(X)<1)=1$, but I cannot obtain $\pi_a^x(X)$ from the definition. Indeed, in the definition of $\pi_a^x(x)= \mathbb{P}(A=a|X=x)$, The two “x”s in $\pi_a^x(x)$ should be mapped to “x” in $\mathbb{P}(A=a|X=x)$. Nevertheless, when $\pi_a^x(x)$ is changed to $\pi_a^x(X)$ or $\pi_a^X(x)$, the mapping procedure is not clear.

**Questions:**

1. According to the definition of $X$, $X=\{X^\emptyset,X^a,X^y,X^\bigtriangleup\}$. At the same time, $X$ is independent of $X^\emptyset$, $X^a$, $X^y$ ,$X^\bigtriangleup$ conditioning to $\Phi(X)$. It is strange to claim Eqn. (4).
2. In the example “Representations with removed noise and instruments”, the authors claim that under Eqn. (5), the validity follows from the d-separation in clustered casual diagram and Appendix B. In appendix B, only investigations related to the example “Invertible representations” are presented.
3. I suspect the equality of $\mathbb{E}(Y[1]-Y[0]|X=x)=\mathbb{E}(Y[1]-Y[0]|X^\bigtriangleup=x^\bigtriangleup, X^y=x^y)$ and $\mathbb{E}(Y[1]-Y[0]|X^\bigtriangleup=x^\bigtriangleup, X^y=x^y)= \mathbb{E}(Y[1]-Y[0]|\Phi(X)=\Phi(x) $ in Eqn. (6) under Eqn. (5). Could the authors provide more details?

---

> ### Author Response · Authors · 2023-11-18
> **Response to Reviewer Pw4L**
>
> We are thankful for your review, and we are happy that you found our paper sound and well-organised. Below, we would like to address the mentioned weaknesses and questions.
>
> **Answer to Weaknesses.** We apologize for the small errors in notation (e.g., in the definitions of $\mu$) and thank you for spotting them. Regarding $\pi_a^x(\cdot)$, **the upper index is not a variable or argument but an indicator** that this propensity relates to the original covariate space (in contrast to $\pi_a^\phi(\cdot)$, which relates to the representation propensity). Therefore, the overlap assumption is defined properly, as only the $x$ inside parenthesis is considered as an argument, namely, $\pi_a^x(X)$. Hence,  $\pi_a^x(\cdot)$ serves as a measurable function, and the probabilistic statement maps a random variable $X$ to the (random) propensity score. Notably, our overlap assumption is equivalent to the $0 < \pi_a^x(x) < 1$ for all $x$ s.t. $\mathbb{P}(X = x) > 0$.
>
> **Action:** We corrected the errors in the revised version of the paper (e.g., we corrected the definitions for $\mu_a^x(x)$ and $\mu_a^\phi(\phi)$). We also clarified the notation around $\pi_a^x(\cdot)$.
>
> **Answers to Questions.**
>
>
>
> 1. Whenever $\Phi(X)$ is invertible wrt. to some sub-covariate, the conditional independence statement contains (partially) random $X$ and deterministic (constant) sub-covariate (due to invertibility). $X$ and its deterministic sub-covariate are then independent by the definition ([https://en.wikipedia.org/wiki/Independence_(probability_theory)#For_real_valued_random_vectors](https://en.wikipedia.org/wiki/Independence_(probability_theory)#For_real_valued_random_vectors)).
>
>     Eq. (4) is a way to formalize that the full information about all four sub-covariates is preserved in the representation. This means that the representation is an invertible function wrt. sub-covariates. Conversely, if the information is partially or fully removed in the representation, the non-independence statement will hold (e.g., in Fig. 1). This notation is e.g. consistent with the literature on the prognostic scores [1].
>
>
>     **Action:** We clarified in the revised version of the paper why Eq. 4 holds.
>
> 2. Thank you for spotting a missing reference.
>
>     **Action:** We added a missing reference to **Lemma 2 (Removal of noise and instruments)** in the revised version of our paper.
>
> 3. Thank you for asking this question. Eq. 6 holds under the assumptions of the clustered causal diagram and invertibility of $\Phi(\cdot)$ wrt. $X^\Delta$ and $X^y$.
>
>     **Action:** We provide the derivation of Eq. 6 in a newly added **Lemma 2 (Removal of noise and instruments)** in a revised version of the paper.
>
>
> **References:**
>
> [1] Ming-Yueh Huang and Kwun Chuen Gary Chan. Joint sufficient dimension reduction and estimation of conditional and average treatment effects. Biometrika, 104(3):583–596, 2017.

---

### Official Review · Reviewer_a9aZ · 2023-11-07

**Soundness:** 4 excellent
**Presentation:** 4 excellent
**Contribution:** 4 excellent
**Rating:** 8
**Confidence:** 4

**Summary:**

This paper addresses the problem of induced confounding that occurs in neural network based conditional average treatment effect estimation as a result of representation learning that operates over a lossy reduced dimension embedding. The authors propose to account for the confounding by leveraging sensitivity analysis. In particular the authors use the marginal sensitivity model and provide bounds on the CATE. A framework is then introduced to estimate the proposed bound within a neural network training flow. A set of experiments are provided which validate the efficacy of the proposed approach.

**Strengths:**

This paper addresses a very important, and often overlooked, aspect of representation learning for causal effect estimation. The authors do a commendable job of describing the circumstances under which we should expect to incur bias due to representation induced confounding, and clearly delineate them from existing approaches which don't suffer from the same issues. The proposed sensitivity analysis is intuitive and the authors do a nice job of describing it's integration into the neural network training process.

**Weaknesses:**

The largest weakness I see is the same as what is commonly shared throughout the sensitivity analysis literature, namely that practitioners must place assumptions on the extent of confounding.

**Questions:**

Given the relative difficulty of CATE estimation in small sample regimes, as the authors point to, it would seem that there are a number of settings where representation based CATE estimation is inappropriate. Given this it would be useful for the authors to compare the bounds provided here and contrast to non-NN based approaches (e.g., BART / causal forests) to give a sense of the relative loss in precision due to the representation induced confounding.

---

> ### Author Response · Authors · 2023-11-18
> **Response to Reviewer a9aZ**
>
> **Answer to Weaknesses.** Thank you. As you nicely point out, a common limitation in the use of MSMs is that the sensitivity parameter (which guides the amount of hidden confounding) must be chosen through expert knowledge. Upon reading your comment, we realized that we should explain more clearly that we do **not** have this limitation but that we can learn the sensitivity parameter from data. The reason is that our framework aims at partial identification of CATE wrt. representations but not sensitivity analysis in the classical sense. Therefore, we employ the marginal sensitivity model (MSM) in an unconventional way. Specifically, we do not require practitioners to pre-specificy the extent of confounding through expert knowledge. Instead, our framework works in a data-driven manner and can infer the sensitivity parameters from the data (see Sec. 4.2, Stage 1). This is a crucial difference between usual MSM applications to detect hidden confounding and our setting where all the confounders are observed.
>
> **Action:** We stress this key difference more clearly in the revised version of the paper. In particular, we highlighted that we do **not** place assumptions on the extent of confounding (see **Section 2**).
>
> **Answer to Questions.** Thank you. This is indeed a great idea, to add classical non-neural CATE estimators for reference. We followed your suggestion and added non-neural CATE  baselines (k-NN, BART, and Causal Forests) to our revised manuscript. These are in our revised **Tables 2-8 and Figures 4-6**. In the same way as for other baselines, we evaluated rPEHE and error rate of policies based on estimated CATE. In the new results, e.g., we observed that non-neural methods outperformed the representation learning CATE estimators on the simple synthetic benchmark with large sample sizes, but fell short in other more complex scenarios, e.g., benchmarks with high-dimensional covariates or low-sample sizes.
>
> **Action:** We added three non-neural CATE baselines, i.e., k-NN, BART, and Causal Forests to the revised version of the paper. See our revised **Tables 2-8 and Figures 4-6.**

---

### Author Response · Authors · 2023-11-18
**Response to all reviewers**

We are grateful for the reviews on our paper and the constructive feedback. We have carefully addressed all of the questions in the individual responses below.

We have incorporated all changes (labelled with **Action**) into the **revised version of our paper**. We have also uploaded the revised version of our paper to OpenOeview. Therein, we highlighted the key changes in **blue font color**.

Our **main improvements** are:

* **New non-neural CATE estimation.** We included several non-neural CATE estimations for comparison (i.e., $k$-NN, BART, and Causal Forest). These are meant as a reference for comparison with the baseline representation learning methods and our framework.
* **More detailed derivations on valid representations and MSM bounds.** We expanded our theoretical analysis and now derive our bounds rigorously (see new **Lemma 3**). We further formalized different properties of our bounds (i.e., sharpness and validity). These are in our new **Corollary 1**.
* **Plots with the ground-truth and estimated decision boundaries.** We added new plots (see new **Figures 7 and 8**) where we show that our estimated bounds are tight enough for practical applications.
* **Better clarifications.** We followed the suggestions of the reviewers and clarified several of our explanations throughout the paper. For example, we clarified (1) how our application of the MSM differs from a traditional application for hidden confounding; (2) that our framework does not assume a specific partitioning of covariates $X$; and (3) the intuition to use conditional interdependencies like in Eq. (4).

We hope that we addressed all concerns of the reviewers. Given our improvements, we are confident that our paper will be a valuable contribution to the causal machine learning literature and a good fit for ICLR 2024.

---

### Meta-Review · Area_Chair_JKJ9 · 2023-12-08

**Metareview:**

The paper provides some new insights on how representation learning interacts with violations of conditional ignorability, and how we could make inference more robust by addressing uncertainty that is due to possible violations of this condition. Overall reviewers found the contribution was original and the problem it addresses is relevant to the substantive community working on representation learning for causal inference research.

**Justification For Why Not Higher Score:**

It *could* go for an oral presentation, although the core problem itself is not extremely original (several other pieces of work exists on the relations between sparsity selection/dimensionality reduction and the problem of effect estimation) The point of view from deep learning though make it particularly suitable for ICLR, so it wouldn't be out of place as an oral presentation. I wouldn't fight for it but I wouldn't oppose it either.

**Justification For Why Not Lower Score:**

The paper raises issues of interest to a non-negligible cohort of researchers working on representation learning for causal inference, so it is good to highlight it for such an audience.

---

### Decision · Program_Chairs · 2024-01-16

Accept (spotlight)